# ReMoE: Fully Differentiable Mixture-of-Experts with ReLU Routing

**Ziteng Wang, Jun Zhu, Jianfei Chen**[*]
Dept. of Comp. Sci. and Tech., Institute for AI, BNRist Center, THBI Lab,
Tsinghua-Bosch Joint ML Center, Tsinghua University
`wangzite23@mails.tsinghua.edu.cn`; `{dcszj,jianfeic}@tsinghua.edu.cn`

## Abstract

Sparsely activated Mixture-of-Experts (MoE) models are widely adopted to scale up model capacity without increasing the computation budget. However, vanilla TopK routers are trained in a discontinuous, non-differentiable way, limiting their performance and scalability. To address this issue, we propose ReMoE, a fully differentiable MoE architecture that offers a simple yet effective drop-in replacement for the conventional TopK+Softmax routing, utilizing ReLU as the router instead. We further propose methods to regulate the router's sparsity while balancing the load among experts. ReMoE's continuous nature enables efficient dynamic allocation of computation across tokens and layers, while also exhibiting domain specialization. Our experiments demonstrate that ReMoE consistently outperforms vanilla TopK-routed MoE across various model sizes, expert counts, and levels of granularity. Furthermore, ReMoE exhibits superior scalability with respect to the number of experts, surpassing traditional MoE architectures. The implementation based on Megatron-LM is available at https://github.com/thu-ml/ReMoE.

## 1 Introduction

Transformer models (Vaswani, 2017) consistently improve performance as the number of parameters increases (Kaplan et al., 2020). However, scaling these models is constrained by computation resources. Sparsely activated Mixture-of-Experts (MoE) (Shazeer et al., 2017) mitigates this challenge by employing a sparse architecture that selectively activates a subset of parameters during both training and inference. This conditional computation allows MoE models to expand model capacity without increasing computational costs, offering a more efficient alternative to dense models.

The key component in MoE is the routing network, which selects the experts to activate for each token. Various routing methods (Shazeer et al., 2017; Lewis et al., 2021; Roller et al., 2021; Zhou et al., 2022) have been proposed, with TopK routing (Shazeer et al., 2017) being the most commonly adopted. However, the vanilla TopK router introduces a discrete and non-differentiable training objective (Shazeer et al., 2017; Zoph et al., 2022), limiting the performance and scalability.

Recent works on fully-differentiable MoE aim to overcome this limitation. Soft MoE (Puigcerver et al., 2023) introduces token merging, while SMEAR (Muqeeth et al., 2023) proposes expert merging. However, both approaches break token causality, making them unsuitable for autoregressive models. Lory (Zhong et al., 2024) improves upon SMEAR and is applicable to autoregressive models. But it underperforms vanilla MoE with TopK routing.

In this work, we address the discontinuities by introducing ReMoE, an MoE architecture that incorporates ReLU routing as a simple yet effective drop-in replacement for TopK routing. Unlike TopK routing, which computes a softmax distribution over the experts and calculates a weighted sum of the largest $K$ experts, ReLU routing directly controls the active state of each expert through a ReLU gate. The number of active experts is determined by the sparsity of the ReLU function. To maintain the desired sparsity, we propose adding a load-balancing refined $L_1$ regularization to the router outputs, with an adaptively tuned coefficient. This approach ensures that ReMoE maintains the same computational costs as TopK-routed MoE.

---

[*]Corresponding author

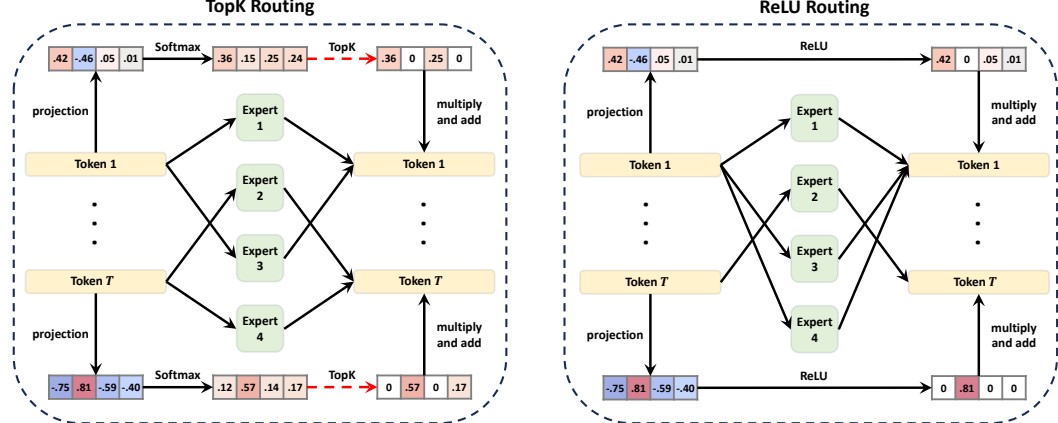

Figure 1: Compute flows of vanilla MoE with TopK routing and ReMoE with ReLU routing. Positive values are shown in orange, and negative values in blue, with deeper colors representing larger absolute values. Zeros, indicating sparsity and computation savings, are shown in white. The red dash arrows in TopK routing indicate discontinuous operations. Compared with TopK routing MoE, ReMoE uses ReLU to make the compute flow fully differentiable.

Compared to TopK routing, ReLU routing is continuous and fully differentiable, as the ReLU function can smoothly transition between zero and non-zero values, indicating inactive and active. Besides, ReLU routing manages the "on/off" state of each expert independently, offering greater flexibility. Moreover, the number of activated experts can vary across tokens and layers, enabling a more efficient allocation of computational resources. Further analysis reveals that ReMoE effectively learns to allocate experts based on token frequency and exhibits stronger domain specialization.

Our experiments on mainstream LLaMA (Touvron et al., 2023) architecture demonstrate that ReLU routing outperforms existing routing methods including TopK routing and fully-differentiable Lory. Through an extensive investigation across model structures, we find that ReMoE consistently outperforms TopK-routed MoE across a broad range of active model sizes (182M to 978M), expert counts (4 to 128), and levels of granularity (1 to 64) (Krajewski et al., 2024). Notably, in terms of scaling behavior, we observe that ReMoE exhibits a steeper performance improvement as the number of experts scales up, surpassing traditional MoE models.

## 2 PRELIMINARIES

### 2.1 MOE FOR DECODER-ONLY TRANSFORMER

A typical decoder-only Transformer model consists of $L$ layers, each containing a Self-Attention module and a Feed-Forward Network (FFN) module. MoE modifies this structure by replacing each FFN module with an MoE module, which comprises a small router and several experts $\text{FFN}_1, \ldots, \text{FFN}_E$, where each expert is equivalent to the original FFN and $E$ denotes the number of experts. Given the input $\boldsymbol{x}^l = (\boldsymbol{x}_t^l)_{t=1}^T \in \mathbb{R}^{T \times d}$ of the layer $l$, where $T$ is the number of tokens in a batch and $d$ is the hidden size, the output $\boldsymbol{y}^l = (\boldsymbol{y}_t^l)_{t=1}^T$ is computed as:

$$\boldsymbol{y}_t^l = \sum_{e=1}^E R(\boldsymbol{x}_t^l)_e \text{FFN}_e(\boldsymbol{x}_t^l; d_{ffn}) \tag{1}$$

Here, $R(\cdot)$ represents the routing function, and $d_{ffn}$ is the intermediate size of the FFN, typically set to $d_{ffn} = 4d$.

### 2.2 TOPK ROUTING

TopK routing (Shazeer et al., 2017; Lepikhin et al., 2020; Fedus et al., 2022) is the most commonly used method for defining the routing function $R(\cdot)$. It introduces sparsity in the MoE computation

by forcibly zeroing out smaller elements:

$$R(\boldsymbol{x}_t^l) = \text{TopK}(\text{Softmax}(\boldsymbol{x}_t^l \boldsymbol{W}_l), k) \qquad (2)$$

where $\boldsymbol{W}_l \in \mathbb{R}^{d \times E}$ is the router's weight matrix, and $\text{TopK}(\cdot, k)$ retains the top $k$ largest values while setting the rest to zero. This mechanism allows for skipping the computation of the $\text{FFN}_e$ functions corresponding to the zeroed-out $R(\boldsymbol{x}_t^l)_e$ values in both the forward and backward passes.

## 3 OUR METHOD: ReMoE

### 3.1 MOTIVATION: FROM TOPK TO RELU

For a given token $\boldsymbol{x} = (x_e)_{e=1}^E$ after Softmax, TopK introduces a jump discontinuity at the $k$-th largest value, denoted as $x_{[k]}$, by zeroing out the values smaller than $x_{[k]}$. This can be expressed as:

$$\text{TopK}(\boldsymbol{x}, k)_e = x_e \cdot \mathbf{1}\{x_e \geq t(\boldsymbol{x}, k)\}, \quad t(\boldsymbol{x}, k) = x_{[k]} \qquad (3)$$

where $\mathbf{1}\{\cdot\}$ is the indicator function, returning 1 if the condition is met and 0 otherwise.

As shown in Figure 2, the jump discontinuity can be eliminated by setting the breakpoint $t(\boldsymbol{x}, k) \equiv 0$, which actually corresponds to the ReLU function:

$$\text{ReLU}(\boldsymbol{x})_e = x_e \cdot \mathbf{1}\{x_e \geq 0\} \qquad (4)$$

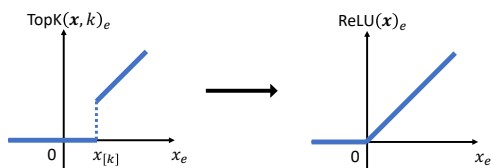

Figure 2: Comparison between TopK and ReLU.

At a high level, ReLU improves upon TopK by aligning the breakpoints of all inputs and setting them to 0. This ensures that the output is continuous at 0, where the experts transition between active and inactive. As a result, the training pipeline becomes fully differentiable.

### 3.2 DIFFERENTIABLE RELU ROUTING

We define the ReLU routing function as follows:

$$R(\boldsymbol{x}_t^l) = \text{ReLU}(\boldsymbol{x}_t^l \boldsymbol{W}_l) \qquad (5)$$

with $(1 - \frac{k}{E})$ being the desired sparsity of ReLU, where $k$ is the number of active experts and $E$ is the total number of experts. This ensures that the computational cost remains equivalent to that of TopK routing.

In vanilla TopK routers, the Softmax outputs sum to 1, representing the probabilities of selecting each expert, after which TopK eliminates those with lower probabilities. In contrast, ReLU routers discard the Softmax function, relying on ReLU's naturally non-negative outputs. The outputs of ReLU routers represent the weights assigned to each expert, which can include 0. Instead of hard-coding expert selection with a discontinuous TopK function, ReLU allows the router to learn which experts to activate (i.e., when to produce 0s) in a fully differentiable manner.

Another key difference is that in TopK routing, each token is routed to exactly $k$ experts, whereas in ReLU routing ReMoE, the routing decisions are independent, allowing tokens to be routed to a variable number of experts. This flexibility is advantageous, as not all tokens have the same level of difficulty. ReMoE can allocate more computational resources to more challenging tokens, a dynamic allocation strategy that we explore further in Section 5.1.

TopK routing introduces a discrete loss function when the set of activated experts changes, whereas ReLU routing remains continuous and fully differentiable. For instance, in a two-expert Top1-routing model, a small weight update that alters the softmax result from $\boldsymbol{x}_1 = (0.51, 0.49)$ to $\boldsymbol{x}_2 = (0.49, 0.51)$ shifts the TopK output from $(0.51, 0)$ to $(0, 0.51)$, creating a discontinuity. In contrast, ReLU routing only changes the activated experts when the routing output is near zero. For example, an output shift from $(0.01, 0)$ to $(0, 0.01)$ remains continuous. Further details on the stability analysis of these two routers can be found in Appendix A.

A comparison of the compute flow between ReMoE and MoE is shown in Figure 1.

### 3.3 CONTROLLING SPARSITY VIA ADAPTIVE $L_1$ REGULARIZATION

ReMoE controls computational costs by managing the sparsity of the ReLU output, targeting a sparsity level of $(1 - \frac{k}{E})$. However, directly training the ReLU router often results in lower sparsity, as the model tends to activate more experts to increase capacity. To meet the desired budget, we need to enforce higher sparsity in the ReLU output.

We achieve this by introducing a regularization loss, $\mathcal{L}_{reg}$, to the loss of language model, $\mathcal{L}_{lm}$:

$$\mathcal{L} = \mathcal{L}_{lm} + \lambda_i \mathcal{L}_{reg}, \tag{6}$$

where $\lambda_i$ is an adaptive coefficient based on the current training step $i$. Initially, we set $\lambda_0$ to a small value and employ a simple zeroth-order algorithm to update it:

$$\lambda_{i+1} = \lambda_i \cdot \alpha^{\text{sign}((1-\frac{k}{E})-S_i)} \tag{7}$$

Here, $\alpha > 1$ is a preset update multiplier, and $S_i$ denotes the average sparsity of all router outputs at the step $i$:

$$S_i = 1 - \frac{1}{LTE} \sum_{l=1}^{L} \sum_{t=1}^{T} \sum_{e=1}^{E} \mathbf{1}\{R(\boldsymbol{x}_t^l)_e > 0\} \tag{8}$$

The idea behind Equation 7 is that when the average sparsity $S_i$ falls below the target sparsity $(1 - \frac{k}{E})$, we increase $\lambda_i$ by a factor of $\alpha$, strengthening the regularization and encouraging higher sparsity. Conversely, if the sparsity exceeds the target, $\lambda_i$ is reduced. We heuristically set $\lambda_0 = 1e^{-8}$ and $\alpha = 1.2$ in all our experiments, and demonstrate the robustness of these hyperparameters in Appendix B.

The regularization term $\mathcal{L}_{reg}$ uses the $L_1$-norm, following prior work (Li et al., 2022; Song et al., 2024), to effectively encourage sparsity:

$$\mathcal{L}_{reg} = \frac{1}{LT} \sum_{l=1}^{L} \sum_{t=1}^{T} \left\| R(\boldsymbol{x}_t^l) \right\|_1 = \frac{1}{LT} \sum_{l=1}^{L} \sum_{t=1}^{T} \sum_{e=1}^{E} R(\boldsymbol{x}_t^l)_e \tag{9}$$

The second equation holds because the output of the ReLU function is non-negative.

The term $\mathcal{L}_{reg}$ represents the average value of all router outputs, including zeros. By taking the derivative of $\lambda_i \mathcal{L}_{reg}$, we observe that the regularization effect adds $\frac{\lambda_i}{LT}$ to the gradient of each non-zero router output, effectively driving the outputs toward zero and enhancing sparsity.

With this $L_1$ regularization, we can control the sparsity around the desired level of $(1 - \frac{k}{E})$ with only minor fluctuations, as shown in Figure 3. Consequently, ReMoE ensures that, on average, tokens are routed to $k$ experts across different layers and tokens, maintaining the same FLOPs as vanilla TopK-routed MoE from a statistical perspective. Our benchmarking results in Appendix D demonstrate that ReMoE can achieve nearly identical training and inference throughputs as conventional MoE, providing an efficient alternative without compromising speed.

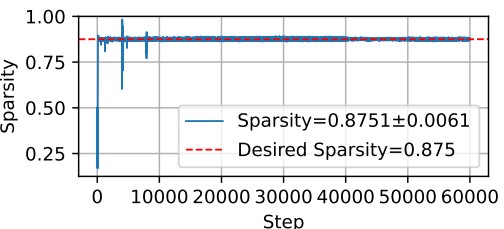

Figure 3: The sparsity of ReMoE with $E = 8, k = 1$ is effectively maintained around the desired target. Sparsity values for all steps are plotted without averaging or sampling. The mean and standard deviation are calculated excluding the first 100 warm-up steps.

### 3.4 INTEGRATE LOAD BALANCING INTO $L_1$ REGULARIZATION

Load imbalance is a significant issue in MoE design, potentially leading to routing collapse (Shazeer et al., 2017; Muennighoff et al., 2024) and uneven computational distribution across multiple devices. The $L_1$ regularization in Equation 9 treats the router output for each expert $e$ and each layer $l$ equally, which can contribute to load balancing problems.

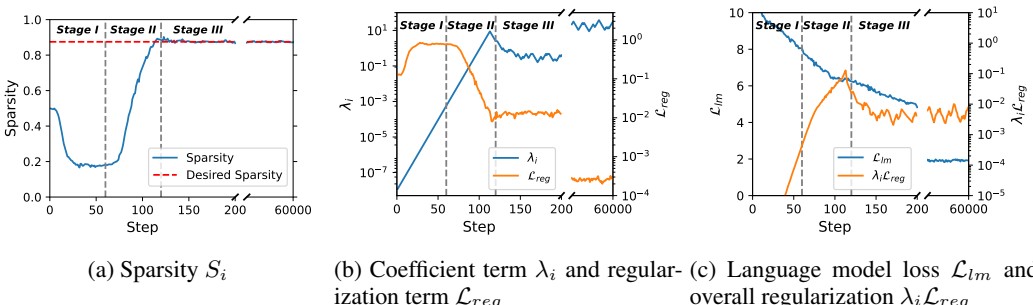

(a) Sparsity $S_i$    (b) Coefficient term $\lambda_i$ and regularization term $\mathcal{L}_{reg}$    (c) Language model loss $\mathcal{L}_{lm}$ and overall regularization $\lambda_i \mathcal{L}_{reg}$

Figure 4: Natural Three Stage Training in ReMoE.

To address this, we introduce a load-balancing refinement to the $L_1$ regularization:

$$\mathcal{L}_{reg,lb} = \frac{1}{LT} \sum_{l=1}^{L} \sum_{t=1}^{T} \sum_{e=1}^{E} f_{l,e} R(\boldsymbol{x}_t^l)_e \tag{10}$$

$$f_{l,e} = \frac{E}{kT} \sum_{t=1}^{T} \mathbf{1}\{R(\boldsymbol{x}_t^l)_e > 0\} \tag{11}$$

Here, $f_{l,e}$ is non-differentiable and represents the average activation ratio of expert $e$ in layer $l$, relative to the desired ratio $\frac{k}{E}$. This serves as a weight for the corresponding router output, modifying the added gradient of non-zero router outputs to $\frac{f_{l,e} \lambda_i}{LT}$. This mechanism penalizes experts receiving more tokens by driving their router outputs toward zero more rapidly.

Although derived from regularization, this formulation is *identical* to the load-balancing loss in vanilla TopK routing (Fedus et al., 2022). In TopK routing, the outputs of Softmax sum to 1, giving the loss a lower bound of 1. In contrast, ReLU routing outputs can be arbitrarily small, making $\mathcal{L}_{reg,lb}$ trivially bounded at 0. Therefore, unlike in MoE, we cannot fix the coefficient $\lambda_i$ in ReMoE, as this would lead to routing collapse toward 0. Thanks to the adaptive update of $\lambda_i$, we can balance sparsity control and load balancing within a single formulation, as given in Equation 10.

Further discussion on load balancing in ReMoE can be found in Section 5.2, and we adopt this load-balancing refined $L_1$ regularization in our later experiments.

### 3.5 NATURAL THREE-STAGE TRAINING IN REMOE

With the regularization scheme described above, we observe a clear and naturally occurring three-stage separation during the training of ReMoE as is depicted in Figure 4.

The first stage is the warm-up stage, or the dense stage. During this stage, $\lambda_i$ is small, while $\mathcal{L}_{lm}$ is large and decreases rapidly. Training ReMoE at this stage is nearly equivalent to training its dense counterpart with the same total number of parameters. Each expert processes more than half of the tokens, allowing the experts to diversify from their random initializations.

The second stage is the sparsifying stage, or the dense to sparse stage. At this point, the sparse regularization term $\lambda_i \mathcal{L}_{reg}$ becomes significant, causing the ReLU routers to activate fewer experts. This forces the experts to become more diverse without causing an increase in $\mathcal{L}_{lm}$.

The third stage is the stable stage, or the sparse stage. In this phase, the sparsity $S_i$ stabilizes at the preset target. During this stage, $\mathcal{L}_{lm}$ is optimized while being softly guided along the sparse subspace by $\mathcal{L}_{reg}$. Both $\mathcal{L}_{reg}$ and $\lambda_i$ change very slowly, with $\mathcal{L}_{reg}$ gradually decreasing and $\lambda_i$ gradually increasing. However, the overall regularization term, $\lambda_i \mathcal{L}_{reg}$, remains relatively constant.

It should be noted that Stages I and II introduce additional computational cost and memory consumption since more experts are activated. However, the time overhead is negligible since they generally require only ∼100 iterations (∼0.17% of the total steps in our setting, benchmarking results are detailed in Appendix D). The memory overhead can be minimized by temporarily reducing the micro-batch size or by employing the activation checkpointing technique that avoids storing intermediate results of activated experts by recomputing them on-the-fly during the backward pass.

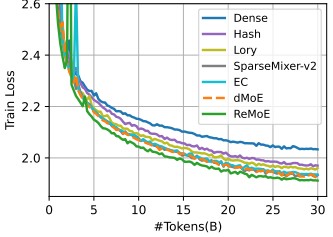

Figure 5: Training curves of different routing methods.

Table 2: Zero-shot accuracy of different routing methods on downstream tasks.

| Model | ARC-c | ARC-e | BoolQ | HellaSwag | LAMBADA | PIQA | RACE | Avg. |
|---|---|---|---|---|---|---|---|---|
| Dense | 19.45 | 43.35 | 54.40 | 28.61 | 31.09 | 61.97 | 28.52 | 38.20 |
| Hash | 19.28 | 45.45 | 54.95 | 29.68 | 31.44 | 63.06 | 27.66 | 38.79 |
| Lory | **20.31** | 42.97 | 49.54 | 28.75 | 32.35 | 62.24 | 27.75 | 37.70 |
| SparseMixer-v2 | 19.80 | **46.72** | 45.96 | 30.24 | 34.12 | 62.89 | 29.00 | 38.39 |
| EC | 18.86 | 42.97 | **60.21** | 29.14 | 29.26 | 61.92 | 27.37 | 38.53 |
| dMoE | 20.05 | 45.16 | 57.83 | 29.83 | 32.97 | **63.55** | 28.33 | 39.67 |
| ReMoE | 20.22 | 46.68 | 54.16 | **30.26** | **35.94** | **63.55** | 29.38 | **40.03** |

## 4 EXPERIMENTS

### 4.1 SETUP

**Infrastructure** We leverage Megatron-LM (Shoeybi et al., 2019) as our code base and implement ReLU routing as a drop-in replacement for the original TopK routing, supporting all forms of model parallelism: Data, Tensor, Pipeline, and Expert Parallelism (Shoeybi et al., 2019; Narayanan et al., 2021; Korthikanti et al., 2023).

**Model Architecture.** We experiment with the mainstream LLaMA (Touvron et al., 2023) architecture, featuring grouped query attention (GQA) (Ainslie et al., 2023), SwiGLU (Shazeer, 2020) activation function, RoPE (Su et al., 2024) position embedding, and RMSNorm (Zhang & Sennrich, 2019). The context length is set to 1024, and the batch size is 512. We experiment with three different dense backbone sizes as shown in Table 1. For vanilla MoE we adopt a load balancing loss of weight 0.01 following Fedus et al. (2022). For ReMoE we use the adaptive load balancing $L_1$ regularization in Equation 10.

**Training Settings.** We train the models on The Pile (Gao et al., 2020), an 800 GB diverse corpus. All models are trained for 60k steps ($\sim$ 30B tokens), which exceeds the compute-optimal dataset size predicted by Krajewski et al. (2024) and is enough to converge. The byte pair encoding (BPE) tokenizer (Sennrich, 2015) is used. We adopt AdamW (Loshchilov, 2017) as the optimizer with $\beta_1 = 0.9, \beta_2 = 0.999$ with ZeRO optimization (Rajbhandari et al., 2020). The learning rate is set to be $5e^{-4}$ with a cosine scheduler. All models are trained with 8 NVIDIA A100 GPUs.

| Size | #Parameters | hidden_size | num_layers | num_heads | num_groups | GFLOPs |
|---|---|---|---|---|---|---|
| Small | 182M | 768 | 12 | 12 | 4 | 995 |
| Medium | 469M | 1024 | 24 | 16 | 4 | 2873 |
| Large | 978M | 1536 | 24 | 16 | 4 | 5991 |

Table 1: Configurations for the dense backbones. FLOPs are calculated with a single sequence according to Narayanan et al. (2021).

### 4.2 COMPARISON WITH OTHER ROUTING METHODS

We compare ReMoE against the following methods: (i) Token-choice dropless TopK routing (dMoE) (Gale et al., 2023) (ii) Expert-choice TopK routing (EC) (Zhou et al., 2022) (iii) Deterministic hash routing (Hash) (Roller et al., 2021) (iv) Fully-differentiable expert-merging routing (Lory) (Zhong et al., 2024) (v) TopK routing with improved gradient estimate (SparseMixer-v2) (Liu et al., 2024b).

The performance of these methods is evaluated with active parameters $N = 182M$ and the expert count $E = 8$. We fix the active expert count to $k = 1$ for straightforward comparison with the dense counterpart. For the Hash method, we use $\mathrm{mod}\ E$ hashing function. And for Lory, the segment length is set to 256, following the original paper.

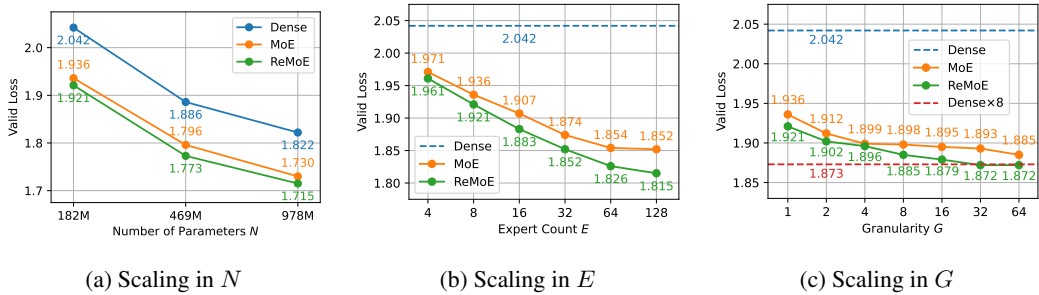

(a) Scaling in $N$        (b) Scaling in $E$        (c) Scaling in $G$

Figure 6: Scalability of ReMoE with respect to the number of active parameters ($N$), expert count ($E$), and granularity ($G$). Default config is $N = 182$M, $E = 8, G = 1, k = 1$. The Y-axis represents the validation loss of each model after training on 30B tokens. ReMoE consistently outperforms MoE across all configurations.

These models are trained on 30B tokens, with the training curves shown in Figure 5, We evaluate the zero-shot performance of the trained models on the following downstream tasks: ARC (Clark et al., 2018); BoolQ (Clark et al., 2019); HellaSwag (Zellers et al., 2019); LAMBADA (Paperno et al., 2016); PIQA (Bisk et al., 2020); RACE (Lai et al., 2017).

The downstream accuracy results are summarized in Table 2.

Our results show that all MoE models outperform the dense model. Deterministic hash routing performs worse than the learned routing methods. Among the Top-K approaches, token-choice dMoE outperforms expert-choice MoE and SparseMixer-v2 in evaluation. The differentiable routing method Lory surpasses Hash routing in training but underperforms in downstream tasks, with both methods falling short of the standard Top-K routing. Notably, ReMoE outperforms all methods, including the mainstream Top-K routing, while benefiting from differentiability.

## 4.3 SCALABILITY OF REMOE

In this section, we compare ReMoE with state-of-the-art dMoE (hereinafter referred to simply as MoE) across varying model parameters $N$, expert counts $E$, and granularity levels $G$ to demonstrate its scalability and universal superiority. Since ReMoE demands more computation in both Stage I and Stage II, we increase the number of training steps for the MoE baseline to match the total computation in each setting, ensuring a more equitable comparison. We present the final validation losses in Figure 6, with comprehensive downstream evaluation results available in Appendix E.

**Scaling in active parameters $N$.** To assess scalability with respect to the number of parameters $N$, we fix $E = 8$ and $k = 1$, while varying active parameters $N$ from 182M to 975M, corresponding to the dense counterpart configurations in Table 1. The total parameters are 777M, 2.58B, 5.73B respectively. The results, shown in Figure 6a, indicate that ReMoE consistently outperforms MoE across all model sizes. The performance gap does not diminish as the model size increases, suggesting that ReMoE maintains its advantage at larger scales.

**Scaling in expert count $E$.** In this experiment, we fix the number of parameters at $N = 182$M and set the number of active experts $k = 1$, while varying the total number of experts $E$ from 4 to 128. The scaling curve in Figure 6b reveals that ReMoE consistently outperforms the standard MoE across all configurations of $E$.

Moreover, a key observation is the steeper slope in ReMoE's performance as $E$ increases, compared to MoE. This suggests that ReMoE scales more effectively with the number of experts and derives greater benefits from larger expert pools. ReMoE's differentiable routing strategy appears better suited for leveraging large expert groups, leading to significant improvements in model expressivity and generalization.

**Scaling in granularity $G$.** We also evaluate ReMoE and MoE in fine-grained settings. Fine-grained MoE (Dai et al., 2024; Krajewski et al., 2024) with granularity $G$ is constructed by dividing

each expert into $G$ smaller experts, as formulated below:

$$\boldsymbol{y}_t^l = \sum_{e=1}^{EG} R(\boldsymbol{x}_t^l)_e \text{FFN}_e(\boldsymbol{x}_t^l; d_{ffn}/G) \tag{12}$$

$$R(\boldsymbol{x}_t^l) = \text{TopK}(\text{Softmax}(\boldsymbol{x}_t^l \boldsymbol{W}_l), kG) \tag{13}$$

Fine-grained MoE outperforms vanilla MoE from a scaling law perspective (Krajewski et al., 2024) and has been adopted in subsequent works (Dai et al., 2024; Tan et al., 2024; Muennighoff et al., 2024). For fine-grained ReMoE, the routing function remains identical to Equation 5, and the target sparsity is still $\left(1 - \frac{k}{E}\right)$. The only distinction lies in the shape of the weight matrix, with $\boldsymbol{W}_l \in \mathbb{R}^{d \times EG}$.

We conduct experiments with $N = 182\text{M}$ and $E = 8$, varying $G$ from 1 to 64 for both fine-grained MoE and fine-grained ReMoE. In addition to comparing these models against the dense baseline with the same number of active parameters, we also evaluate their dense counterpart with the same total number of parameters. This is achieved by expanding the intermediate size of the FFN by a factor of $E$, which we denote as *Dense×8*. This configuration represents the strict upper bound for MoE and ReMoE, as it is equivalent to a Mixture-of-Experts with all experts activated (Dai et al., 2024).

As illustrated in Figure 6c, fine-grained ReMoE consistently outperforms fine-grained MoE. Moreover, fine-grained ReMoE of $G = 32$ and $G = 64$ reach the performance of the theoretical upper bound, *Dense×8*, while requiring significantly fewer FLOPs during both training and inference. In contrast, fine-grained MoE is unable to match in all settings, making ReMoE a more efficient and effective choice.

## 5 DISCUSSION

### 5.1 DYNAMIC EXPERT ALLOCATION IN REMOE

In ReMoE, each token dynamically activates a subset of experts, allowing the model to adaptively allocate resources. We evaluate the performance of the $N = 182\text{M}, E = 8, k = 1$ ReMoE model and analyze the relationship between token frequency and the average number of active experts. As illustrated in Figure 7, the model tends to assign a higher number of experts to rarer tokens, such as '©', 'OTAL', and '@#', while reducing the number of active experts for more frequent tokens like ' ', '\n', and 'the'.

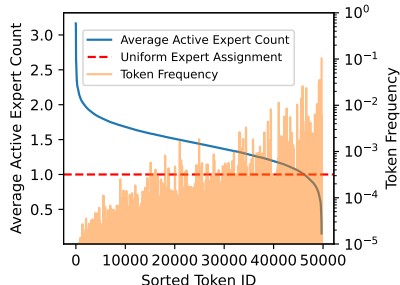

Figure 7: Correlation between expert allocation and token frequency in ReMoE. X-axis is sorted by average active expert count and token frequency is in log-scale.

This adaptive behavior mirrors the principles of a Huffman tree Huffman (1952), where more frequent symbols are assigned shorter codes, and rarer symbols are assigned longer codes. Similarly, ReMoE tends to "cluster on" common tokens by activating fewer experts, effectively compressing the "representation" of these frequent tokens. In contrast, for rarer tokens, ReMoE activates a more diverse set of experts, "encoding" them as a richer linear combination at the expert level. This suggests that ReMoE learns to dynamically allocate computational resources, achieving an efficient balance between resource usage and the model's capacity, optimizing performance under a constrained expert budget. Dynamic expert allocation is also evident at the domain level, as detailed in Appendix G.

### 5.2 THE ROLE OF LOAD BALANCING IN REMOE

Load imbalance can lead to routing collapse in the vanilla TopK-routed MoE, where the router tends to assign the same expert to all inputs, in which scenario the training objective becomes continuous and fully differentiable. As is shown in Figure 8a, there is a significant performance gap between MoE models with and without load balancing (LB).

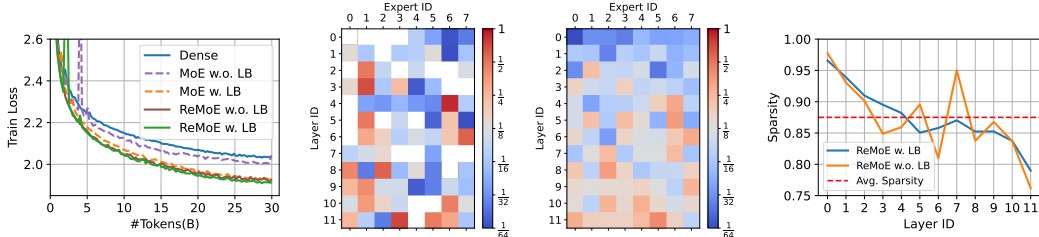

(a) Training curves of MoE and ReMoE with and without load balancing

(b) Average routed tokens ratio of ReMoE w.o. LB

(c) Average routed tokens ratio of ReMoE w. LB

(d) Sparsity across different layers in ReMoE

Figure 8: Observations on the role of load balancing in MoE and ReMoE. White squares in (b) represent inactive experts with fewer than 1/64 tokens routed to them.

While in ReLU routing, thanks to its differentiablity, even applying the $L_1$ regularization from Equation 9 without load balancing yields comparable results with a well-tuned MoE with LB. However, some experts in ReMoE without LB remain inactive, illustrated as white squares in Figure 8b which shows the heat map of the *average routed tokens ratio* (i.e., the fraction of tokens routed to the $e$-th expert in the $l$-th layer) over 50M tokens in test set. This inactivity can limit the model's capacity.

When load balancing is incorporated into the refined $L_1$ regularization (Equation 10), the experiments show a more even distribution of token assignments across experts, with all experts being utilized, as shown in Figure 8c. The final loss in ReMoE decreases after introducing load balancing.

Besides, we observe ReMoE with LB can produce a smoother sparsity distribution across layers as depicted in Figure 8d. This is because $f_{l,e}$ is computed based on the absolute number of routed tokens, meaning denser layers receive stronger penalties.

Note that even ReMoE with load balancing (LB) does not yield a perfectly even distribution. However, the trade-off between load balancing and performance can be easily adjusted by modifying the $L_1$ regularization in Equation 10. For instance, changing $f_{l,e}$ to $f_{l,e}^2$ would make the model more sensitive to load imbalance. Additionally, device-level load balancing techniques, as proposed in Dai et al. (2024), could also be employed. Since load imbalance in ReMoE does not lead to severe routing collapse, it primarily becomes a hardware utilization issue. As such, we leave the exploration of these variants for future work.

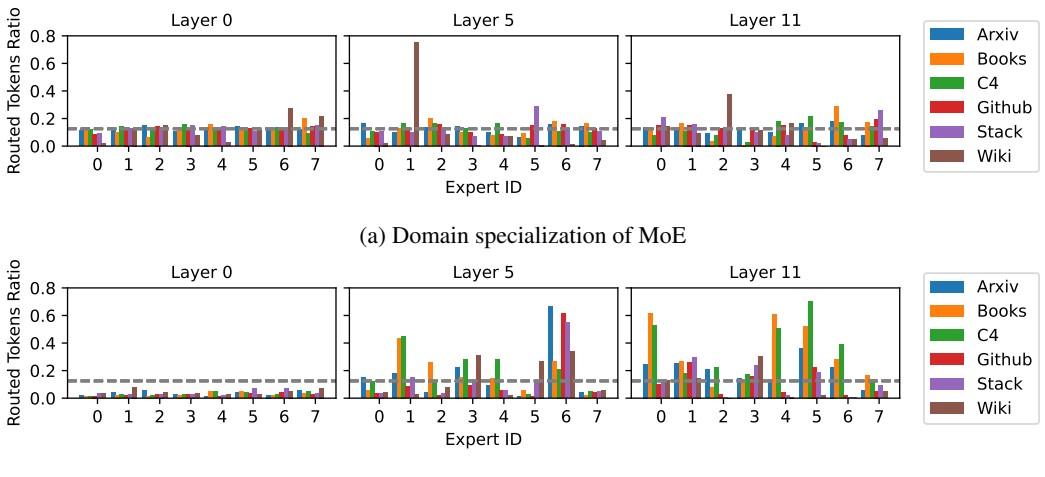

Figure 9: Average routed tokens ratio for MoE and ReMoE across 12 layers and 8 experts in different domains. The gray dashed lines indicate uniform distribution. ReMoE shows stronger domain specialization.

## 5.3 Domain Specialization in ReMoE

The differentiability and dynamic allocation strategy of ReMoE facilitates the development of diverse experts that specialize in different domains. This allows the router to effectively perform ensemble learning by leveraging the expertise of various experts, as demonstrated in our experiments.

In Figure 9, we plot the average routed tokens ratio across different experts, layers, and domains—namely Arxiv, Books, C4, Github, Stackexchange, and Wikipedia—for MoE and ReMoE models with $N = 182M, E = 8$. We focus on the first, middle, and last layers (with IDs 0, 5, and 11). The results for most experts in MoE (Figure 9a) show a roughly uniform distribution across all domains. In contrast, experts in ReMoE (Figure 9b) exhibit clear domain specialization, being activated with varying frequencies across different domains. For example, more than half of the tokens from Arxiv, Github, and StackExchange—domains that emphasize structured, non-natural languages like LaTeX and Python—are routed to Expert 6 in Layer 5, significantly more than in other domains. A more detailed result of domain specialization can be found in Appendix F.

# 6 Related Works

## 6.1 Mixture-of-Experts

Mixture-of-Experts (MoE) was initially proposed in the early 1990s (Jacobs et al., 1991; Jordan & Jacobs, 1994) and later introduced into large-scale neural networks as a sparse submodule for efficiency (Shazeer et al., 2017). Advances like GShard (Lepikhin et al., 2020) and Switch Transformer (Fedus et al., 2022) integrated sparse MoE into Transformer models, achieving significant results. More recently, MoE has been used in commercial-scale language models such as Mixtral-8x7B (Jiang et al., 2024), DeepSeekMoE 16B (Dai et al., 2024), and Snowflake Arctic 17B (Snowflake, 2024).

## 6.2 Routing Mechanisms in MoE

Various routing methods have been developed for expert selection. Static routers, such as BASE (Lewis et al., 2021), use predefined rules like combinatorial optimization, while Hash routing (Roller et al., 2021) relies on deterministic hash functions, and THOR (Zuo et al., 2021) assigns experts randomly with regularization. Learned routers adaptively select experts based on token input, using approaches like REINFORCE (Bengio et al., 2013; Schulman et al., 2015; Clark et al., 2022) for reinforcement learning, and TopK routing (Shazeer et al., 2017; Zhou et al., 2022) for token or expert selection, though TopK introduces discontinuities that hinder gradient estimation.

## 6.3 Differentiable Mixture-of-Experts

Recent work on fully differentiable MoE models addresses the challenges of discrete optimization, basically through token merging and expert merging approaches. Soft MoE (Puigcerver et al., 2023) uses token merging, assigning fixed slots to each expert as a linear combination of input tokens. SMEAR (Muqeeth et al., 2023) merges experts into an ensemble via weighted averaging. However, both methods require a full probability map of input tokens, making them unsuitable for autoregressive models. Lory (Zhong et al., 2024) preserves autoregressiveness by segmenting sentences to merge experts but underperforms compared to TopK routing.

# 7 Conclusion

In this paper, we propose ReMoE, a fully differentiable MoE architecture with ReLU routing. The simple yet effective ReLU routing function acts as a drop-in replacement for the conventional TopK+Softmax routing, offering (i) continuity and differentiability, and (ii) dynamic expert allocation across tokens and layers. With the adaptive load balancing $L_1$ regularization, ReMoE universally outperforms TopK-routed MoE across various model sizes, expert counts, and levels of granularity, demonstrating sharper performance gains as the number of experts scales.

ACKNOWLEDGMENT

The authors gratefully acknowledge Chao Du and Tianyu Pang for the insightful discussions. This work was supported by the NSFC Project (No. 62376131), Tsinghua Institute for Guo Qiang, and the High Performance Computing Center, Tsinghua University. J.Z is also supported by the XPlorer Prize.

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

# A  STABILITY ANALYSIS OF TOPK AND RELU

We introduce two metrics, "flip rate" and "flip count", to evaluate the routing stability:

$$\text{flip rate} = \frac{\sum_{l=1}^{L} \left\| \text{vec}(\boldsymbol{M}_i^l - \boldsymbol{M}_{i-1}^l) \right\|_1}{LTE} \tag{14}$$

$$\text{flip count} = E \times \text{flip rate} \tag{15}$$

where $\boldsymbol{M}_i^l \in \mathbb{R}^{T \times E}$ denotes the 0-1 mask matrix of the output of the router at layer $l$ and training step $i$, computed using a *fixed* calibration set of tokens.

The metric "flip rate" represents the percentage of expert activation states that change (from active to inactive or conversely) in a single update, while "flip count" indicates the average number of experts whose activation states change.

We measure the two metrics on MoE and ReMoE with $N =$182M and $E \in \{8, 16, 32\}$ training for 10B tokens. The results are presented in Figure 10, indicating that the ReLU router is more stable than the TopK router:

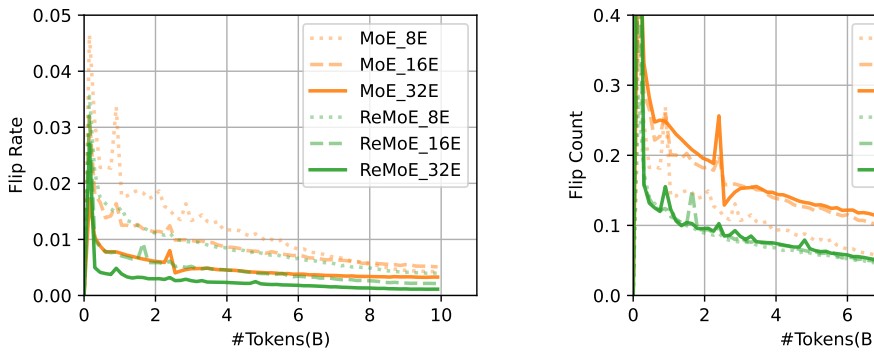

Figure 10: Flip rate and flip count of MoE and ReMoE

When $E = 8$, we find the flip rate of MoE is higher than ReMoE, though the gap narrows as training progresses and the learning rate decreases. While for $E = 16$ and $E = 32$, the flip rate of MoE remains consistently $2 - 3\times$ higher compared to ReMoE throughout training.

Moreover, the flip count of ReMoE is invariant with respect to $E$, whereas the flip count of MoE is highly sensitive to the total number of experts and keeps increasing as $E$ grows.

Notably, the flips in TopK-routed MoE are discontinuous (e.g.$(0.51, 0) \to (0, 0.51)$), while those in ReLU-routed ReMoE are continuous(e.g.$(0.01, 0) \to (0, 0.01)$), further underscoring the superiority of the ReLU router.

# B  INSENSITIVITY TO $\lambda_0$ AND $\alpha$

| $\lambda_0$ | $1e^{-16}$ | $1e^{-12}$ | $1e^{-8}$ | $1e^{-4}$ | 1 |
|---|---|---|---|---|---|
| Valid Loss | 2.031 | 2.029 | 2.032 | 2.036 | 2.032 |
| Settling time | 138 | 136 | 110 | 55 | 92[†] |

[†] Overshoot observed in 8-92 steps.

Table 3: Valid loss and settling time for different values of $\lambda_0$ with $\alpha = 1.2$.

| $\alpha$ | 1.05 | 1.1 | 1.2 | 1.3 | 1.5 |
|---|---|---|---|---|---|
| Valid Loss | 2.033 | 2.028 | 2.032 | 2.029 | 2.057[*] |
| Settling time | 414 | 211 | 110 | 80 | 52 |

[*] A large oscillation amplitude in sparsity is observed.

Table 4: Valid loss and settling time for different values of $\alpha$ with $\lambda_0 = 1e^{-8}$.

The ReMoE adaptation algorithm in Equation 7 includes two hyperparameters: $\lambda_0$ and $\alpha$. *Settling time*, defined as the total number of steps required in Stage I and Stage II (as outlined in Section 3.5),

is influenced by these parameters. For all experiments, we set $\lambda_0 = 1e^{-8}$ and $\alpha = 1.2$, but we show that performance remains stable as long as $\lambda_0$ is small and $\alpha$ is close to 1.

Our experiments with $N = 182M$, $E = 8$, $G = 1$, and $k = 1$ ReMoE models trained for 20k steps (∼10B tokens) reveal only minor variations in validation loss for different $\lambda_0$ values (Table 3) and $\alpha$ values (Table 4), except for $\alpha = 1.5$ which caused rapid regularization changes and excessive oscillation. Besides, although different $\lambda_0$ and $\alpha$ values affect settling time, the impact is minor compared to the overall training steps, proving the insensitivity.

## C  PERFORMANCE FOR LONGER TRAINING

We conduct experiments of training MoE and ReMoE for a longer duration. We experiment with $N =$ 469M, $E = 8$, $k = 1$ and train the models with a batch size of 4M tokens and training over 120B tokens. The results, as shown in Table 5, indicate that the superiority of ReMoE persists in longer training.

| Model | Valid Loss | ARC-c | ARC-e | BoolQ | HellaSwag | LAMBADA | PIQA | RACE | Avg. |
|---|---|---|---|---|---|---|---|---|---|
| MoE | 1.716 | 23.62 | 52.40 | 53.94 | 35.43 | 43.64 | 68.34 | **31.48** | 44.12 |
| ReMoE | **1.689** | **25.34** | **55.22** | **55.96** | **36.76** | **45.82** | **68.93** | 30.43 | **45.49** |

Table 5: Performance of training $N =$ 469M, $E = 8$, $k = 1$ models for 120B tokens.

## D  SPEED COMPARISON OF REMOE AND MOE

We measure the end-to-end training time for MoE and ReMoE with models of $N =$ 469M training over 120B tokens. The time consumption across stages is summarized in Table 6. We find Stage I and Stage II account for ∼1.02% of the total training time and incur ∼0.58% overhead.

| Model | Stage I | Stage II | Stage III | Total |
|---|---|---|---|---|
| MoE | 0.12 | 0.41 | 119.12 | 119.65 |
| ReMoE | 0.32 | 0.91 | 119.25 | 120.48 |

Table 6: End-to-end training time comparison across stages (in hours). The time is measured on $N = 469M$, $E = 8$, $k = 1$ models training over 120B tokens.

| # Parameters | TP | Model | Train TFLOPS | Train Diff. | Infer TFLOPS | Infer Diff. |
|---|---|---|---|---|---|---|
| 182M | 1 | MoE | 103.49 | ↑1.82% | 78.47 | ↑2.19% |
| | | ReMoE | 105.38 | | 80.19 | |
| 469M | 1 | MoE | 138.58 | ↓1.37% | 107.52 | ↑3.89% |
| | | ReMoE | 136.69 | | 111.71 | |
| 978M | 1 | MoE | 160.46 | ↓1.77% | 153.11 | ↓0.23% |
| | | ReMoE | 157.61 | | 152.76 | |
| 978M | 2 | MoE | 133.40 | ↓0.68% | 118.55 | ↓1.08% |
| | | ReMoE | 132.49 | | 117.27 | |
| 978M | 4 | MoE | 103.61 | ↓2.29% | 85.96 | ↑2.33% |
| | | ReMoE | 101.23 | | 87.96 | |

Table 7: Throughput comparison between TopK-routed MoE and ReLU-routed ReMoE models. TP indicates the tensor parallel size. Train Diff. and Infer Diff. indicate the relative TFLOPS difference of ReMoE compared to MoE, where ↑ denotes ReMoE is faster, and ↓ denotes it is slower.

We further measure the throughput of ReMoE against TopK-routed MoE across different model sizes and tensor parallel sizes during Stage III. The results, presented in Table 7, indicate that ReMoE achieves comparable training and inference speeds with MoE, with a minor deviation ranging from $-2.29\%$ to $+3.89\%$. This speed consistency is desirable, as ReMoE introduces only a minimal modification to the standard MoE architecture by adjusting the routing function, thereby avoiding additional computational overhead.

## E  DOWNSTREAM EVALUATION RESULTS

This section provides the detailed downstream evaluation results for the main experiments of scalability of ReMoE in Section 4.3 and ablations on load balancing in Section 5.2.

### E.1  SCALING IN ACTIVE PARAMETERS $N$

The downstream evaluation results for scaling with respect to the parameter count $N$, as discussed in Section 4.3, are presented in Table 8. These results highlight the performance comparison with increasing model parameters.

| Model | $N$ | ARC-c | ARC-e | BoolQ | HellaSwag | LAMBADA | PIQA | RACE | Avg. |
|---|---|---|---|---|---|---|---|---|---|
| Dense | 182M | 19.45 | 43.35 | 54.40 | 28.61 | 31.09 | 61.97 | 28.52 | 38.20 |
| | 469M | 21.50 | 49.12 | 56.88 | 31.12 | 36.74 | 64.47 | 30.53 | 41.48 |
| | 978M | 21.93 | 50.88 | **60.24** | 32.42 | 41.06 | 67.46 | **31.77** | 43.68 |
| MoE | 182M | 20.82 | 45.03 | 57.55 | 29.84 | 31.81 | 63.28 | 28.42 | 39.53 |
| | 469M | 23.63 | 52.40 | 53.94 | 32.43 | 43.64 | 68.34 | 31.48 | 43.69 |
| | 978M | 23.81 | 52.90 | 58.90 | 35.01 | **44.42** | 67.90 | 31.48 | 44.91 |
| ReMoE | 182M | 20.22 | 46.68 | 54.16 | 30.26 | 35.94 | 63.55 | 29.38 | 40.03 |
| | 469M | 21.67 | 53.16 | 58.75 | 33.80 | 40.66 | 67.95 | 31.20 | 43.88 |
| | 978M | **24.06** | **55.26** | 57.28 | **35.93** | **44.42** | **68.99** | 30.43 | **45.20** |

Table 8: Downstream results of scaling in active parameters $N$.

### E.2  SCALING IN EXPERT COUNT $E$

Table 9 contains the downstream evaluation results for scaling with respect to the expert count $E$, as examined in Section 4.3. This analysis illustrates how varying the number of experts influences the overall model effectiveness of MoE and ReMoE.

| Model | $E$ | ARC-c | ARC-e | BoolQ | HellaSwag | LAMBADA | PIQA | RACE | Avg. |
|---|---|---|---|---|---|---|---|---|---|
| Dense | - | 19.45 | 43.35 | 54.40 | 28.61 | 31.09 | 61.97 | 28.52 | 38.20 |
| MoE | 4 | 20.73 | 44.49 | 59.63 | 29.14 | 31.40 | 63.33 | 29.19 | 39.70 |
| | 8 | 20.82 | 45.03 | 57.55 | 29.84 | 31.81 | 63.28 | 28.42 | 39.53 |
| | 16 | 20.90 | 45.29 | 46.36 | 30.50 | 33.22 | 64.96 | 28.33 | 38.50 |
| | 32 | 19.54 | 47.35 | 52.29 | 31.12 | 35.63 | 64.25 | 28.23 | 39.77 |
| | 64 | 19.88 | 46.63 | **60.06** | 31.47 | 36.33 | 65.07 | 28.04 | 41.06 |
| | 128 | **20.99** | 47.69 | 56.73 | 32.00 | 36.62 | 65.67 | 28.04 | 41.10 |
| ReMoE | 4 | 19.88 | 46.46 | 57.43 | 29.64 | 33.57 | 62.95 | 27.66 | 39.66 |
| | 8 | 20.22 | 46.68 | 54.16 | 30.26 | 35.94 | 63.55 | 29.38 | 40.03 |
| | 16 | 20.90 | 49.28 | 53.36 | 30.85 | 37.09 | 65.83 | **30.05** | 41.05 |
| | 32 | 20.56 | 48.11 | 59.54 | 31.42 | 37.84 | 65.18 | 28.42 | 41.58 |
| | 64 | 20.82 | 50.51 | 57.80 | 32.17 | 36.74 | 65.78 | 27.46 | 41.61 |
| | 128 | 19.97 | **51.05** | 56.97 | **32.40** | **37.92** | **66.70** | 29.86 | **42.12** |

Table 9: Downstream results of scaling in expert count $E$.

## E.3 SCALING IN GRANULARITY $G$

The downstream evaluation results for scaling with respect to the granularity $G$ are shown in Table 10, based on the experiments in Section 4.3. These results demonstrate the superiority of fine-grained ReMoE over fine-grained MoE.

| Model | $G$ | ARC-c | ARC-e | BoolQ | HellaSwag | LAMBADA | PIQA | RACE | Avg. |
|-------|-----|-------|-------|-------|-----------|---------|------|------|------|
| Dense | - | 19.45 | 43.35 | 54.40 | 28.61 | 31.09 | 61.97 | 28.52 | 38.20 |
| Dense×8 | - | **22.78** | 48.11 | 59.66 | 31.11 | 35.65 | 65.02 | 29.57 | **41.70** |
| | 1 | 20.82 | 45.03 | 57.55 | 29.84 | 31.81 | 63.28 | 28.42 | 39.53 |
| | 2 | 21.42 | 46.55 | 54.25 | 29.95 | 32.52 | 64.09 | 28.61 | 39.62 |
| | 4 | 20.99 | 46.09 | 55.90 | 30.52 | 35.16 | 63.98 | 29.28 | 40.27 |
| MoE | 8 | 21.59 | 47.73 | 60.70 | 30.83 | 36.41 | 64.69 | 28.04 | 41.42 |
| | 16 | 19.80 | 48.82 | 57.34 | 30.64 | 36.00 | 64.74 | 28.71 | 40.86 |
| | 32 | 21.67 | 48.78 | 57.85 | 31.27 | **37.10** | 64.69 | 28.52 | 41.41 |
| | 64 | 20.14 | 48.74 | **61.50** | 31.03 | 36.31 | 63.93 | 27.85 | 41.35 |
| | 1 | 20.22 | 46.68 | 54.16 | 30.26 | 35.94 | 63.55 | 29.38 | 40.03 |
| | 2 | 20.14 | 47.39 | 57.95 | 30.60 | 34.52 | 63.71 | 28.52 | 40.40 |
| | 4 | 20.39 | 47.94 | 55.35 | 31.04 | 36.11 | 64.64 | 29.00 | 40.64 |
| ReMoE | 8 | 20.82 | 48.36 | 60.49 | 30.90 | 36.06 | 63.87 | 28.90 | 41.34 |
| | 16 | 21.25 | **49.41** | 56.06 | 30.91 | 36.23 | 64.91 | 29.95 | 41.25 |
| | 32 | 20.90 | 48.86 | 55.81 | 31.14 | 36.58 | 64.69 | **30.05** | 41.15 |
| | 64 | 20.65 | 48.74 | 60.06 | **31.56** | 36.43 | **65.40** | 29.00 | 41.69 |

Table 10: Downstream results of scaling in granularity $G$.

## E.4 LOAD BALANCING ABLATIONS

Table 11 presents the downstream evaluation results for the load balancing ablations, as discussed in Section 5.2. These results compare performance with and without load balancing, offering insights into the different roles of load balancing in MoE and ReMoE.

| Model | LB | ARC-c | ARC-e | BoolQ | HellaSwag | LAMBADA | PIQA | RACE | Avg. |
|-------|-----|-------|-------|-------|-----------|---------|------|------|------|
| Dense | - | 19.45 | 43.35 | 54.40 | 28.61 | 31.09 | 61.97 | 28.52 | 38.20 |
| MoE | × | 19.20 | 44.74 | 50.80 | 28.60 | 30.18 | 62.24 | 27.94 | 37.67 |
| MoE | ✓ | 20.05 | 45.16 | **57.83** | 29.83 | 32.97 | **63.55** | 28.33 | 39.67 |
| ReMoE | × | 19.45 | 46.34 | 56.94 | 30.19 | 31.79 | 63.33 | 28.61 | 39.52 |
| ReMoE | ✓ | **20.22** | **46.68** | 54.16 | **30.26** | **35.94** | **63.55** | **29.38** | **40.03** |

Table 11: Downstream results of training with or without load balancing.

# F DETAILED RESULTS FOR DOMAIN SPECIFICATION

Figure 11 shows the average routed tokens ratio of MoE and ReMoE across all layers. ReMoE demonstrates significantly stronger domain specialization compared to MoE, where certain experts are more frequently activated for specific domains. This suggests that ReMoE is better at learning and exploiting the unique characteristics of different domains, allowing it to allocate computational resources more effectively. In contrast, MoE exhibits a more uniform expert activation across domains, indicating less differentiation in its expert specialization.

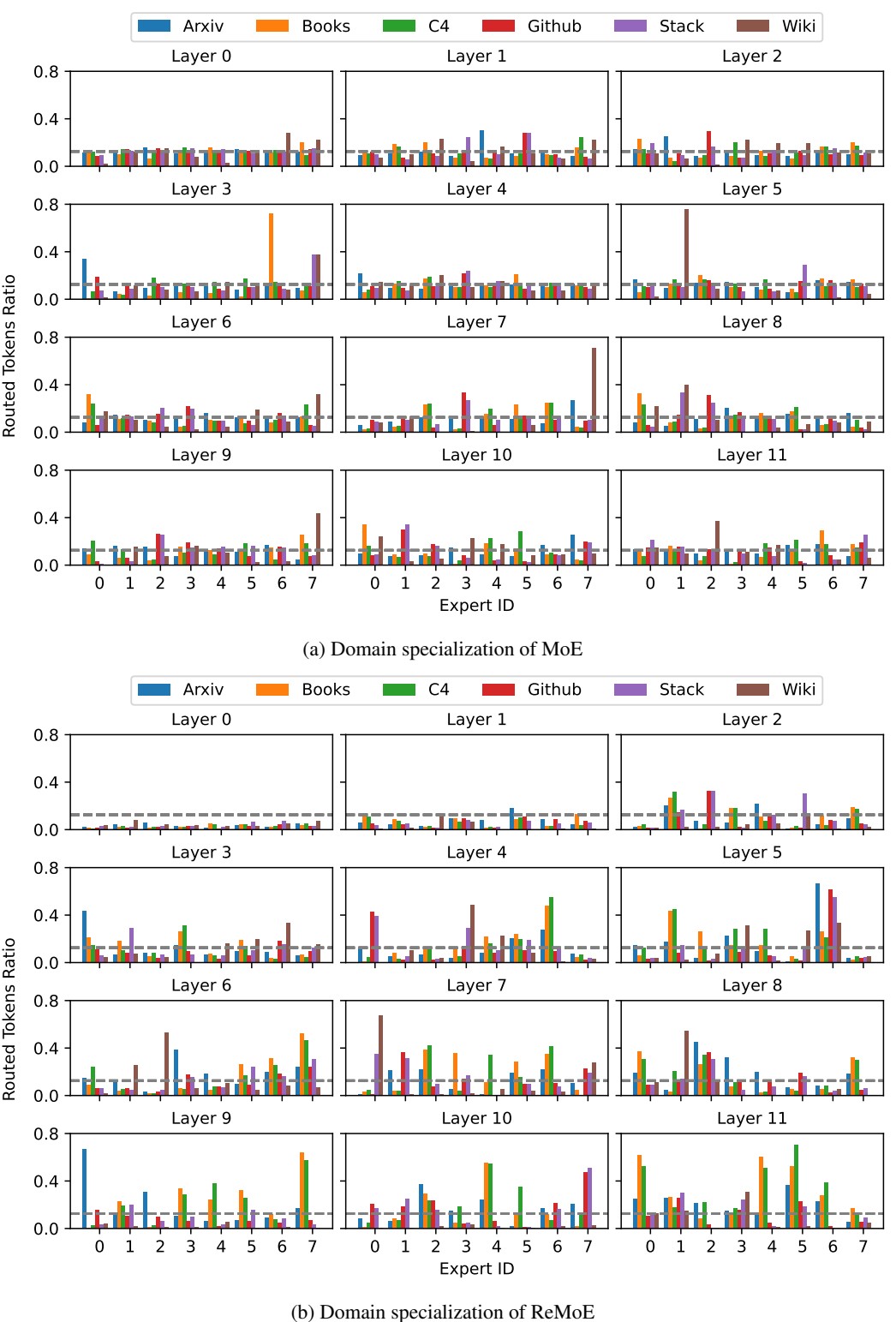

(a) Domain specialization of MoE

(b) Domain specialization of ReMoE

Figure 11: Detailed results of average routed tokens ratio for MoE and ReMoE in different domains.

We further analyze the experts in Layer 5 of ReMoE and observe that certain highly related, domain-specific vocabularies are consistently routed to the same expert. To investigate this, we calculate the routing probabilities of different tokens based on their IDs, defined as the ratio of the number of times a specific expert is utilized to the total occurrences of the token. The results are summarized in Table 12.

Our findings reveal that the vocabularies exhibit clear specialization, reflecting domain-specific characteristics. For example, Expert 1, which is more frequently assigned to natural language domains (e.g., Books, C4), tends to route tokens such as `husband`, `wife`, and `lover`. In contrast, Expert 6, which is associated with non-natural language domains (e.g., Arxiv, Github, StackExchange), predominantly routes code-related tokens like `variable`, `env`, and `HEAD`.

| Expert ID | Routed Tokens With High Probability |
|---|---|
| 0 | `End`(100%); `folding`(100%); `Fill`(100%); `FILE`(100%); `NULL`(100%); `byte`(100%); `Release`(99.36%); `Del`(99.80%) |
| 1 | `husband`(100%); `ife`(100%); `baby`(100%); `human`(100%); `lover`(99.60%); `).`(99.86%); `),`(99.71%); `)...`(98.425%) |
| 2 | `invest`(100%); `Fortune`(100%); `exec`(100%); `0000`(100%); `Sorry`(100%); `bye`(97.82%); `If`(97.74%); `®`(97.63%) |
| 3 | `Conversely`(100%); `Methods`(100%); `flower`(100%); `Blossom`(99.93%); `Argentina`(100%); `Georgian`(100%); `Uruguay`(98.90%); `African`(100%) |
| 4 | `Spring`(100%); `Summer`(100%) `Autumn`(100%); `Winter`(100%); `seasons`(99.02%); `Temperature`(100%); `hot`(97.98%); `cold`(100%) |
| 5 | `è`(100%); `æ`(99.80%); `å`(98.59%); `Æ`(97.67%) |
| 6 | `]);`(100%); `gif`(100%); `size`(100%); `variable`(100%); `env`(100%); `begin`(97.95%); `HEAD`(97.94%); `|`(97.83%) |
| 7 | `Kuala`(100%); `Tus`(100%); `Lama`(100%); `Riley`(98.94%) |

Table 12: Routed tokens with high probability for experts in Layer 5 of ReMoE

## G    DOMAIN-LEVEL DYNAMIC EXPERT ALLOCATION IN ReMoE

We measure the average active expert count across different domains, as shown in Figure 12, and find that the computation allocation in ReMoE also varies at the domain level. Furthermore, this variation increases in deeper layers closer to the output. This is reasonable because deeper layers tend to capture more abstract and domain-specific features, leading to more pronounced specialization in expert activation.

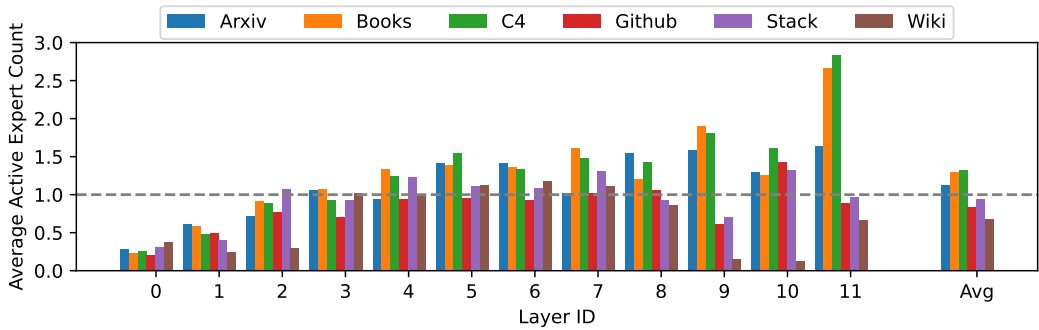

Figure 12: Domain-level dynamic expert allocation

## H    TRAINING MoE WITH NEAR-DENSE WARMUP

In ReMoE, the training process naturally progresses through three stages, with the first two involving near-dense training where the majority of experts are active. To facilitate a fairer comparison, in Section 4.3, we train the MoE model for additional tokens to match the overall computational cost. In this section, we explore an alternative approach by introducing a similar near-dense warmup phase for MoE, referred to as "MoE with warmup," to align its computational footprint with ReMoE across each stage. Specifically, we train the MoE with $N = 182M$, $E = 8$, and $k = 6$—approximately matching the average sparsity of ReMoE during Stages I and II, as depicted in Figure 4a—for the first 100 steps, before transitioning to $k = 1$ for the remainder of the training process.

Table 13 compares this warmup variant to both standard MoE and ReMoE. The results indicate that the warmup phase provides a modest improvement in validation loss compared to standard MoE, despite matching the overall computational cost. Nonetheless, ReMoE consistently outperforms both variants. This suggests that the three-stage training pipeline learned by ReMoE, with Stages I and II comprising only the first 100 steps, is beneficial to overall performance.

| Model | Valid Loss | ARC-c | ARC-e | BoolQ | Hella-Swag | LAM-BADA | PIQA | RACE | Avg. |
|---|---|---|---|---|---|---|---|---|---|
| MoE | 1.936 | 20.82 | 45.03 | 57.55 | 29.84 | 31.81 | 63.28 | 28.42 | 39.53 |
| MoE with warmup | 1.928 | 20.73 | 46.38 | 52.35 | 30.28 | 33.90 | 63.76 | 27.66 | 39.29 |
| ReMoE | 1.921 | 20.22 | 46.68 | 54.16 | 30.26 | 35.94 | 63.55 | 29.38 | 40.03 |

Table 13: Performance of MoE with near-dense warmup

We further extend our experiments with MoE using warmup to configurations with larger $E$, which increases the computational cost of near-dense training. The results, summarized in Table 14, show that as $E$ increases, the warmup setting consistently improves performance. However, ReMoE still outperforms both variants, maintaining a steeper performance scaling with respect to $E$.

| Model, $E =8$ | Valid Loss | Avg. Acc. | Model, $E =32$ | Valid Loss | Avg. Acc. | Model, $E =128$ | Valid Loss | Avg. Acc. |
|---|---|---|---|---|---|---|---|---|
| MoE | 1.936 | 39.53 | MoE | 1.874 | 39.77 | MoE | 1.852 | 41.10 |
| MoE with warmup | 1.928 | 39.29 | MoE with warmup | 1.869 | 40.06 | MoE with warmup | 1.841 | 41.34 |
| ReMoE | 1.921 | 40.03 | ReMoE | 1.852 | 41.58 | ReMoE | 1.815 | 42.12 |

Table 14: Results for MoE with warmup under different expert count $E$

To further investigate the impact of warmup steps on MoE performance, we vary the number of warmup steps for the $E = 8$ MoE configuration among 50, 100, 500, and 1000. The training curves of these models, along with standard MoE and ReMoE, are shown in Figure 13, and the final validation losses are summarized in Table 15.

Our results reveal that performance does not improve monotonically with an increasing number of warmup steps, despite the additional computation. This behavior arises due to the discrepancy between the training objectives of $k = 6$ (warmup phase) and $k = 1$ (post-warmup phase). For instance, when warmup concludes after 100 steps, the transition between phases is smooth, with the loss changing minimally from $6.491 \rightarrow 6.751$. However, extending warmup to 500 or 1000 steps leads to a more pronounced loss gap of $3.101 \rightarrow 5.827$ and $2.695 \rightarrow 4.428$, respectively.

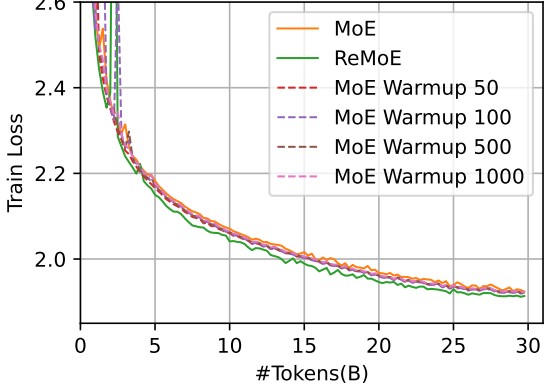

| Model | Warmup Steps | Valid Loss |
|-------|:---:|:---:|
| MoE | 0 | 1.937 |
|  | 50 | 1.930 |
|  | 100 | 1.928 |
|  | 500 | 1.930 |
|  | 1000 | 1.931 |
| ReMoE | - | 1.921 |

Table 15: Final validation loss of MoE with different warmup steps

Figure 13: Training curves of MoE with different warmup steps

In summary, near-dense warmup can enhance the performance of TopK MoE when training from scratch by providing a better initialization for the experts. However, the warmup phase should conclude while the language model loss is still decreasing rapidly. Prolonging the warmup can exacerbate the gap between the warmup and subsequent training phases, ultimately degrading performance. In contrast, ReMoE naturally determines the appropriate warmup steps and sparsity levels due to its continuous and differentiable training dynamics.

## I  FUTURE DIRECTIONS

This work can be advanced in the following ways:

- **ReLU Routing for Mixture-of-LoRAs (MoLoRA).** MoLoRA (Zadouri et al., 2023; Wu et al., 2024; Jiao et al., 2024) integrates MoE architectures to manage multiple Low-Rank Adaptation (LoRA) experts, dynamically activating task-specific adapters during inference. ReMoE's fully differentiable routing mechanism could enhance MoLoRA by enabling smoother transitions between LoRA experts, particularly when adapters are trained on diverse tasks. Using ReLU straightforwardly in MoLoRA is explored in RoDE (Jiao et al., 2024), which can be further enhanced by scaling the expert count while controlling the sparsity as in ReMoE.

- **ReLU Routing in Product-Key-Memory (PKM) Networks.** PKM (Lample et al., 2019; He, 2024; Berges et al., 2024; Huang et al., 2024) architectures treat individual neurons as ultra-fine-grained experts, leading to routing complexity at unprecedented scales (e.g., millions of experts). ReMoE's differentiable routing and steep scaling properties are particularly suited to address PKM's optimization challenges.

- **Synergy with Efficient Attention Algorithms.** Merging ReMoE's sparse, conditional feed-forward computation with efficient attention variants—such as quantized (Zhang et al., 2024b;a), linearized (Sun et al., 2023; Gu & Dao, 2023), sparse (Jiang et al., 2025; Gao et al., 2024), or mixture-of-attention (Zhang et al., 2022; Csordás et al., 2025) mechanisms—could enable Transformers to scale efficiently in both sequence length and model capacity without incurring additional computational overhead.

- **Dynamic Expert Pruning for ReMoE.** ReMoE's differentiable training inherently promotes expert specialization, with significant variance in expert importance across domains. This property makes ReMoE more amenable to expert pruning (Lu et al., 2024; Liu et al., 2024a) compared to traditional TopK-routed MoE architectures.

