# OpenReview forum: "ReMoE: Fully Differentiable Mixture-of-Experts with ReLU Routing"
_ICLR.cc/2025/Conference — ICLR 2025 Poster_

### Official Review · Reviewer_rQKz · 2024-11-03

**Soundness:** 3
**Presentation:** 3
**Contribution:** 3
**Rating:** 8
**Confidence:** 4

**Summary:**

This paper proposes a new routing method for MoE routing which addresses issues related to discontinuity inherent in TopK routing while still maintaining sparsity. The key idea is to replace TopK softmax with a ReLU routing function. This leads to a natural selection of experts from the non-zero routing scores. A sparse selection of experts is achieved via L1-regularization which evolves dynamically throughout training in order to eventually reach a target sparsity. The L1-regularization can also be combined with a load balancing loss to ensure both sparse selection of experts as well as an even distribution of token routing.

**Strengths:**

Mixture-of-experts models are a powerful new paradigm which can unlock favorable scaling. However, training such models is difficult, in large part due to the discrete nature of the TopK expert routing. The proposed routing method alleviates this difficult which enables better capabilities in MoE models. The nature of this solution is conceptually clean and gives good results. The paper is well-written and provides a useful empirical analysis.

**Weaknesses:**

It would be helpful to delve a bit more into the conceptual intuitions of the regularization penalty, especially with the additional load balancing and the dynamic penalty adjustment. This will make it easier for the reader to grasp the key conceptual contribution. Additionally, it would be helpful to shed some light on exactly what issue is being overcome with the new strategy.

**Questions:**

Why is TopK routing struggling beyond just "being discontinuous" and how is the new strategy overcoming this? Is it really the continuity of the routing or the more flexible nature of the sparse regularization which allows for multiple "phases" of training?

---

> ### Author Response · Authors · 2024-11-22
> **Response to Reviewer rQKz**
>
> We greatly thank the reviewer for recognizing our method as **conceptually clean** and for acknowledging that it produces **good results**. We sincerely hope that our responses will further clarify our contributions and address any remaining uncertainties. Below, we provide detailed replies to the reviewer’s questions:
>
> **Q1: Conceptual Intuitions on Regularization**
>
> Thank you for the thoughtful advice! We will clarify the conceptual intuitions for $L_1$ regularization, load balancing, and dynamic adjustment as follows:
>
> - **$L_1$ Regularization:** This introduces a force to each non-zero router output by adding $\frac{\lambda_i}{LT}$ to their gradients, driving the outputs toward zero (inactive) and enhancing sparsity.
> - **Load Balancing:** By adding a weight $f_{l,e}$ to the regularization, the added gradient of non-zero router outputs is modified to $\frac{f_{l,e} \lambda_i}{L T}$, which depends for each expert. This penalizes over-utilized experts by driving their outputs toward zero more quickly.
> - **Dynamic Adjustment:** The coefficient $\lambda_i$ dynamically adjusts the magnitude of the regularization force applied to router outputs, ensuring the model stabilizes at the target sparsity level.
>
> We have incorporated these conceptual intuitions into Sections 3.3 and 3.4 to clarify the narrative further.
>
> **Q2: What issue is being overcome with ReLU routing**
>
> The superiority of ReLU over TopK can be summarized into the following two aspects:
> - **Continuity and differentiability**: ReLU is fully differentiable and stays continuous when the active experts change, where TopK fails.
> - **Dynamic expert allocation**: ReLU fertilizes dynamic expert allocation, so that tokens are not necessarily routed to $k$ experts and more computations are allocated to rarer tokens.
>
> We have modified the Conclusion Section to highlight the above statements.
>
> **Q3: Why is TopK routing struggling beyond just "being discontinuous" and how is ReLU overcoming it?**
>
> - Theoretically, the discontinuity in TopK causes the router to receive false gradient when active experts change. For example, consider a simple MoE $f(x)=\tilde\theta_1f_1(x)+\tilde\theta_2f_2(x)$, where $\tilde{\theta_1},\tilde{\theta_2}=\text{Top1}(\theta_1,\theta_2)$. Depending on the comparison, we have $f(x)=\theta_1f_1(x)$ if $\theta_1>\theta_2$ and $f(x)=\theta_2f_2(x)$ if $\theta_1<\theta_2$. The output $f(x)$ is discontinuous and thus non-differentiable at $\theta_1=\theta_2$ since $f_1\ne f_2$. When active expert changes from $f_1$ to $f_2$, the sparse gradient computed with $f_1$​ no longer accurately reflects the true gradient for the router, which is now related to $f_2$​. This issue does not arise with ReLU, as its gradient is inherently sparse and aligned with its output.
> - Empirically, we further measured the percentage of expert activation states that change in a single update on a calibration set ("flip rate" of the router outputs). Our results show that the flip rate of the TopK router is consistently higher than that of the ReLU router, with the gap increasing as $E$ grows (up to $2-3\times$ more flips for TopK at $E=32$). This indicates that the ReLU router produces more stable routing decisions. For more details, please refer to Appendix A.
>
> **Table: Flip rate comparison with $N=$ 182M, $E=32$, $k=1$**
>
> | Model | \#Tokens=2B | \#Tokens=4B | \#Tokens=6B | \#Tokens=8B | \#Tokens=10B |
> | ----- | ----------- | ----------- | ----------- | ----------- | ------------ |
> | MoE   | 0.0059      | 0.0045      | 0.0035      | 0.0034      | 0.0034       |
> | ReMoE | 0.0030      | 0.0025      | 0.0017      | 0.0013      | 0.0011       |
>
>
>
> **Q4: Attribution of the success: continuity, dynamic expert allocation or multiple phases of training.**
>
> Thank the reviewer for the insightful question!
>
> The multiple phases of training is not critical to the performance improvement, as Stage I/II takes only ~0.17% of the total training and Stage III is just the same with TopK. But it does provide a stable initialization to the experts of ReMoE and is indisposable.
>
> As for the other two improvements—continuity and dynamic expert allocation—it is challenging to determine which contributes more significantly to the effectiveness of the ReLU router. These two aspects are inherently intertwined in the ReLU solution: a continuous and fully differentiable router ensures continuity at zero for each router output, allowing the outputs to transition smoothly between zero and non-zero independently, which leads to dynamic expert allocation. Conversely, dynamic expert allocation suggests that the router can flexibly update the activation states of experts, which inherently requires a degree of smoothness around zero, thereby ensuring continuity. Therefore, both continuity and dynamic expert allocation are essential to ReLU router, working together to enhance its effectiveness.

---

> > ### Comment · Reviewer_rQKz · 2024-11-24
> >
> > Thank you for the replies. I will keep my current rating.

---

> > > ### Author Response · Authors · 2024-11-25
> > > **Thank You for Your Support**
> > >
> > > Dear Reviewer rQKz,
> > >
> > > Thank you for your kind support of our work! We believe that ReLU-routed ReMoE represents a meaningful progression from the current discontinuous TopK-routed MoE and hope it inspires future research on fully differentiable Mixture-of-Experts.
> > >
> > > Best regards,
> > >
> > > The Authors

---

### Official Review · Reviewer_yJh9 · 2024-11-04

**Soundness:** 4
**Presentation:** 4
**Contribution:** 3
**Rating:** 6
**Confidence:** 5

**Summary:**

Traditional Mixture of Experts (MoE) models use Top-K routing to select experts, which is a non-differentiable process. In this work, the authors propose using ReLU activation combined with load balancing loss and sparsity loss to address this issue. Several validation experiments are conducted to demonstrate the effectiveness of the proposed approach.

**Strengths:**

- The authors propose a simple, intuitive, and straightforward alternative method for MoE routing, which can be integrated into existing MoE structures via a drop-in approach.
- The submission clearly illustrates the background information and the proposed method.
- The performance gain by replacing the traditional routing with ReLU is interesting.
- The submission provides many insightful observation, for instance, the analysis on the correlation between expert allocation and token frequency, and the effect of load balance loss.

**Weaknesses:**

- The experiment scale is relatively small, and only perplexity is reported. I'm wondering about the performance comparison on downstream tasks.
- The performance improvement seems significant, but it would help if the authors can provide more analysis and experiment/theoritical evidence on why the fully differentiable router will help.

**Questions:**

- What does $N$ refers to? If I understand correctly, $N$ refers to the number of activated parameters. In that case, the largest model should have about 7/8 billion parameters in total, right? I suggest the authors to explicitly mention the important hyper-parameter like the model size.
- The Figure 7 is really interesting and insightful. But I'm wondering whether the router will assign different number of experts solely based on token ids (or token frequency), or it will also capture some semantic informaton (i.e. difficulty of tasks).

---

> ### Author Response · Authors · 2024-11-22
> **Response to Reviewer yJh9**
>
> We thank the reviewer for considering our work as **interesting** and **insightful** and for the valuable feedback. We hope that our responses provide additional clarity. Below we address the concerns:
>
> **Q1: Small experiment scale**
> - Model scale: The largest model in our experiments is the 978M × 8 model, comprising a total of 5.7B parameters.
> - Data scale: We extend the longest training duration from 30B tokens to 120B tokens for the 469M $\times$ 8 model with total parameters of 2.5B in Appendix C. The results demonstrate that ReMoE continues to outperform MoE:
>
> **Table: Results for active parameters $N=$ 469M  and total parameters of 2.5B models trained on 120B tokens**
>
> | Model | Valid Loss | Downstream Avg. Acc. | ARC-c | ARC-e | BoolQ | HellaSwag | LAMBADA | PIQA  | RACE  |
> | ----- | ---------- | -------------------- | ----- | ----- | ----- | --------- | ------- | ----- | ----- |
> | MoE   | 1.716      | 44.12                | 23.62 | 52.40 | 53.94 | 35.43     | 43.64   | 68.34 | 31.48 |
> | ReMoE | 1.689      | 45.49                | 25.34 | 55.22 | 55.96 | 36.76     | 45.82   | 68.93 | 30.43 |
>
>
>
> **Q2: Downstream evaluation results:**
>
> In the revised version, we have included downstream evaluation results for **all experiments**. These results demonstrate that ReMoE consistently outperforms MoE in downstream tasks, aligning with the perplexity results. For a detailed breakdown of the results, please refer to Appendix E.
>
> **Q3: Analysis on why fully differentiable ReLU router will help**
>
> The ReLU router helps mainly in the following two aspects:
> - **Differentiability:** In TopK, expert activation state switches are discontinuous (e.g.$(0.51,0)\rightarrow(0,0.51)$, even though the pre-TopK outputs transition as $(0.51,0.49)\rightarrow(0.49,0.51)$)In contrast, the ReLU router provides continuous and differentiable transitions (e.g.$(0.01,0)\rightarrow(0,0.01)$), ensuring smooth gradient flow.
> - **Dynamism:** The ReLU router allows tokens to dynamically route to a variable number of experts, which can be learned by the model. In comparison, TopK enforces the selection of exactly $k$ experts per token, regardless of differences in token importance.
>
> To further analyze this, we measure the percentage of expert activation states that change in a single update on a calibration set ("flip rate" of the router outputs). Our results show that the flip rate of the TopK router is consistently higher than that of the ReLU router, with the gap increasing as $E$ grows (up to $2-3\times$ more flips for TopK at $E=32$). This indicates that the ReLU router produces more stable routing decisions. For more details, please refer to Appendix A.
>
> **Table: Flip rate comparison with $N=$ 182M, $E=32$, $k=1$**
>
> | Model | \#Tokens=2B | \#Tokens=4B | \#Tokens=6B | \#Tokens=8B | \#Tokens=10B |
> | ----- | ----------- | ----------- | ----------- | ----------- | ------------ |
> | MoE   | 0.0059      | 0.0045      | 0.0035      | 0.0034      | 0.0034       |
> | ReMoE | 0.0030      | 0.0025      | 0.0017      | 0.0013      | 0.0011       |
>
>
> **Q4: Meaning of $N$**
>
> We apologize for the confusion. $N$ refers to the active parameters. The largest model of 978M $\times$ 8 has 5.7B total parameters.  We have updated the subtitle "Scaling in parameters $N$" to "Scaling in active parameters $N$" and added the total parameters count to the main paragraph in Section 4.3 to avoid misunderstandings.
>
> **Q5: Dynamic allocation based on semantic information**
>
> We sincerely thank the reviewer for the insightful question! To inspect this, we measured the number of activated experts across different domains, with the results presented in Appendix G. Our findings show that ReMoE effectively learns to allocate computations dynamically based on semantic information. It assigns varying numbers of experts to different domains, with these differences becoming more pronounced in the deeper layers. Thanks again for your advice!
>
> **Table: Average active expert count of ReMoE on different domains:**
>
> | Domain | Layer 0 | Layer 5 | Layer 11 | Avg.   |
> | ------ | ------- | ------- | -------- | ------ |
> | Arxiv  | 0.2789  | 1.4121  | 1.6358   | 1.1262 |
> | Books  | 0.2266  | 1.3927  | 2.6665   | 1.2942 |
> | C4     | 0.2515  | 1.5505  | 2.8319   | 1.3208 |
> | Github | 0.1996  | 0.9584  | 0.8938   | 0.8335 |
> | Stack  | 0.3090  | 1.1061  | 0.9691   | 0.9453 |
> |

---

> ### Author Response · Authors · 2024-11-25
> **Looking Forward to Your Feedback**
>
> Dear Reviewer yJh9:
>
> We really appreciate your insightful and constructive comments that help us improve this paper. We have taken our maximum effort to address all your concerns.
>
> As the ICLR public discussion phase concludes in two days, we kindly ask if our responses have fully addressed your concerns. If there are any remaining issues, we’d be happy to provide further clarifications. If you feel that your concerns have been resolved, we would greatly appreciate it if you could consider raising the score.
>
> Thank you once again for your time and thoughtful input. We hope you have a wonderful day!
>
> Best wishes,
>
> The Authors

---

> > ### Comment · Reviewer_yJh9 · 2024-11-25
> >
> > Dear authors,
> >
> > Thanks for your reply. I will keep my original score.
> >
> > Best,

---

> > > ### Author Response · Authors · 2024-11-25
> > > **Thank You for Your Support**
> > >
> > > Dear Reviewer yJh9,
> > >
> > > Thank you for your kind support of our work! We believe that ReLU-routed ReMoE represents a meaningful progression from the current discontinuous TopK-routed MoE and hope it inspires future research on fully differentiable Mixture-of-Experts.
> > >
> > > Best regards,
> > >
> > > The Authors

---

### Official Review · Reviewer_URhd · 2024-11-04

**Soundness:** 3
**Presentation:** 3
**Contribution:** 2
**Rating:** 5
**Confidence:** 4

**Summary:**

The paper introduce ReMoE, a differentiable MoE architecture that incorporates ReLU routing in replacement for TopK routing for Mixture-of-Experts. This paper further propose methods to regulate the router’s sparsity while balancing the load among experts. ReMoE enables efficient dynamic allocation of computation across tokens and layers, exhibiting domain specialization. The paper observes a natural three-stage training process in ReMoE: warm-up stage, sparsifying stage, and stable stage offering interesting insights. Perplexity based experiments demonstrate that ReMoE consistently outperforms vanilla TopK-routed MoE across various model sizes, expert counts, and levels of granularity.

**Strengths:**

1. The paper is well-written and authors have presented some interesting ablations like domain specialization of experts, assign a higher number of experts to rarer tokens, etc.

2. Experimental results presented looks promising. It is interesting to see how a relatively simple idea work so much better.

3. Three-stage training that naturally occurs during REMoE training is an interesting section. I recommend author to add the loss plots too in the figure to draw parallels.

**Weaknesses:**

While there are many interesting experiments and ablation in the paper, I have several comments.

1. The authors have failed to provide the training cost of different baselines used in the experiments and also for inference. Clearly, ReMoE stage I training activates a lot more experts during initial stage. Some speed analysis is must to clearly identify the computation overhead of dynamic expert selection both from inference and training perspective. A more detailed analysis of the trade-offs between performance gains and computational costs would be beneficial.

2. Complete evaluation setup of paper is centered around training and validation loss, and perplexity which is completely not reliable to justify performance. To illustrate the performance benefit of ReMoE, it is critically important to show results translating to downstream tasks.

3. TopK routing facilitates better parallelization opportunities while scaling MoEs. What are the authors thoughts on adopting a dynamic expert selection which can lead to poor GPU utilization.

4. The authors introduces two new hyperparameters $\lambda_{0}$ and $\alpha$. MoE training is well-known to be tricky. I am concerned how these hyperparameters will further increase challenges. The authors have indeed provided some ablation on 20k
steps (∼10B tokens) for 182M token, but it is not clear how the observations replicate in large MoEs. In addition, even for this small-scale experiment, some $\alpha$ values lead to rapid regularization changes and excessive oscillation.

5. While the paper compares ReMoE with several baselines, it would be beneficial to include comparisons with other recent MoE variants, such as those addressing the routing bottleneck or improving expert utilization.

6. Domain specialization experiments are interesting. I would recommend authors to conduct some data domain vocabulary coverage ratio related experiments for each experts to complete the full-picture.

I am looking forward to additional experiments in rebuttal to update my rating.

**Questions:**

See above.

---

> ### Author Response · Authors · 2024-11-22
> **Response to Reviewer URhd (1/2)**
>
> We thank the reviewer for recognizing our work as **interesting** and **promising** and for offering thoughtful and constructive feedback. We sincerely hope that our responses can provide additional clarity and would greatly appreciate it if the reviewer could reconsider the score. Below we address the reviewer’s concerns in detail:
>
> **Q1: Training speed analysis**
>
> We have added a detailed analysis of training speed for MoE and ReMoE in Appendix D. This includes an end-to-end training time comparison with a breakdown across the three stages, and throughput measurements for both training and inference under varying active parameters and parallelism configurations. The key results are briefly summarized in the following two aspects:
>
> - Overhead of Stages I/II: Stages I and II do introduce additional computational overhead due to activating more experts. However, these stages account for only 0.17% of the overall training process, resulting in a negligible overhead of just **0.58%**:
>
> **Table: End-to-End Training Time Comparison of MoE and ReMoE (Unit: Hours).**
>
> | Model | Stage I/II | Stage III | Total  |
> | ----- | ---------- | --------- | ------ |
> | MoE   | 0.53       | 119.12    | 119.65 |
> | ReMoE | 1.23       | 119.25    | 120.48 |
>
>
> - Overhead of dynamic expert selection in ReMoE: the dynamic expert selection in ReMoE doesn't introduce overhead with state-of-the-art MoE implementation of grouped-GEMM. We don't trade speed for performance: the training and inference speeds of ReMoE and MoE are comparable:
>
> **Table: relative throughput of ReMoE over MoE.**
>
> | N    | Training | Inference |
> | ---- | -------- | --------- |
> | 182M | ↑1.82%   | ↑2.19%    |
> | 469M | ↓1.37%   | ↑3.89%    |
> | 978M | ↓1.77%   | ↓0.23%    |
>
>
> **Q2: Downstream evaluation results**
>
> In the revised version, we have included downstream evaluation results for **all experiments**. These results demonstrate that ReMoE consistently outperforms MoE in downstream tasks, aligning with the perplexity results. For a detailed breakdown of the results, please refer to Appendix E.
>
> **Q3: GPU utilization concern of dynamic expert selection:**
>
> In practical MoE computation, an all-to-all dispatch and combine mechanism is employed, grouping tokens routed to the same expert. As a result, the number of experts chosen per token is inconsequential, as the tokens are ultimately combined into input matrix blocks for each expert with variable shapes.
>
> For ReMoE, we implement a similar all-to-all token dispatcher as in TopK MoE with capacity constraints. Like ReMoE, TopK MoE with capacity also results in uneven expert selection for tokens, where tokens are not routed to exactly K experts. This does not introduce GPU utilization issues.
>
> Empirically, we measure the throughput of both training and inference of ReMoE and MoE and find they are comparable in speed, as stated in Q1.
>
> **Q4: Concern on introduced hyperparameters:**
>
> We would like to emphasize that across all our experiments, spanning various model sizes, expert counts, and granularities, we consistently used **the same** set of heuristically preset hyperparameters, specifically $\alpha=1.2,\lambda_0=1e^{-8}$, without any additional tuning. This consistency demonstrates the robustness of the hyperparameters.
>
> Regarding the oscillation, the presented case of $\alpha = 1.5$ was intended as an abnormal value in our ablation study. With $\alpha = 1.5$, the regularization coefficient changes by 1.5 times with each update, representing an unusually large step that leads to oscillations. However, for more reasonable $\alpha$ values close to 1—specifically in the range of 1.05 to 1.3—the validation loss remains stable and consistent.
>
> To further validate robustness, we replicated the ablation studies on larger models with active parameters $N =$ 469M and total parameters of 2.5B. The results confirm that the performance remains robust across hyperparameter settings. We hope these results can alleviate the reviewer's concern.
>
> **Table: Robustness of introduced hyperparameters on ReMoE with $N=$ 469M active parameters.**
>
> | $\lambda_0$   | $1e^{-12}$ | $1e^{-8}$ | $1e^{-4}$ | \|  | $\alpha$      | $1.1$ | $1.2$ | $1.3$ |
> | ------------- | ---------- | --------- | --------- | --- | ------------- | ----- | ----- | ----- |
> | Settling Time | 124        | 101       | 50        | \|  | Settling Time | 187   | 101   | 74    |
> | Valid Loss    | 1.886      | 1.880     | 1.883     | \|  | Valid Loss    | 1.886 | 1.880 | 1.888 |

---

> > ### Author Response · Authors · 2024-11-22
> > **Response to Reviewer URhd (2/2)**
> >
> > **Q5: Comparison with other recent MoE variants**
> >
> > We further compare ReMoE with SparseMixer-v2 \[1\], a method proposed September 2024 by Microsoft to improve the router's gradient estimate. We find ReMoE outperforms SparseMixer-v2. The results are put in Section 4.2 and summarized below:
> >
> > **Table: Comparison of ReMoE with SparseMixer-v2**
> >
> > | Model          | Valid Loss | Downstream Avg. Acc. | ARC-c | ARC-e | BoolQ | HellaSwag | LAMBADA | PIQA  | RACE  |
> > | -------------- | ---------- | -------------------- | ----- | ----- | ----- | --------- | ------- | ----- | ----- |
> > | MoE            | 1.937      | 39.67                | 20.05 | 45.16 | 57.83 | 29.83     | 32.97   | 63.55 | 28.33 |
> > | SparseMixer-v2 | 1.935      | 38.39                | 19.80 | 46.72 | 45.96 | 30.24     | 34.12   | 62.89 | 29.00 |
> > | ReMoE          | 1.921      | 40.03                | 20.22 | 46.68 | 54.16 | 30.26     | 35.94   | 63.55 | 29.38 |
> >
> >
> > **Q6: Domain vocabulary coverage ratio**
> >
> > Thank you for the constructive question! We further analyzed tokens with high routing probabilities for different experts and observed that specialization in domain vocabularies aligns closely with domain specialization. We have included this analysis in Appendix F. Please refer to it for detailed information.
> >
> > **Table: Partial results for specialization in domain vocabularies**
> >
> > | Expert ID | Specialized Domain           | Routed Tokens With High Probability                                                                                     |
> > | --------- | ---------------------------- | ----------------------------------------------------------------------------------------------------------------------- |
> > | 2         | Books, C4                    | `husband`(100%); `ife`(100%); `baby`(100%); `human`(100%); `lover`(99.60%); `).`(99.86%); `),`(99.71%); `)...`(98.425%) |
> > | 6         | Arxiv, Github, StackExchange | `]);`(100%); `gif`(100%); `size`(100%); `variable`(100%); `env`(100%);<br>`begin`(97.95%); `HEAD`(97.94%); `\|`(97.83%) |
> >
> >
> >
> > \[1\] GRIN: GRadient-INformed MoE https://arxiv.org/pdf/2409.12136

---

> ### Author Response · Authors · 2024-11-25
> **Looking Forward to Your Feedback**
>
> Dear Reviewer URhd:
>
> We really appreciate your insightful and constructive comments that help us improve this paper. We have taken our maximum effort to address all your concerns.
>
> As the ICLR public discussion phase concludes in two days, we kindly ask if our responses have fully addressed your concerns. If there are any remaining issues, we’d be happy to provide further clarifications. If you feel that your concerns have been resolved, we would greatly appreciate it if you could consider raising the score.
>
> Thank you once again for your time and thoughtful input. We hope you have a wonderful day!
>
> Best wishes,
>
> The Authors

---

> ### Author Response · Authors · 2024-12-01
> **Looking Forward to Further Discussion**
>
> Dear Reviewer URhd,
>
> We sincerely appreciate the time and effort you have dedicated to reviewing our paper, as well as your thoughtful feedback and recognition of our work. We have carefully addressed your questions to the best of our ability.
>
> As the extended discussion period deadline approaches, please do not hesitate to reach out if you have any additional questions or require further clarification. We would be delighted to continue the conversation and address any remaining concerns.
>
> Thank you once again for your valuable insights!
>
> Best wishes,
> The Authors

---

### Official Review · Reviewer_qrQT · 2024-11-04

**Soundness:** 4
**Presentation:** 3
**Contribution:** 3
**Rating:** 8
**Confidence:** 5

**Summary:**

The paper proposes a new routing method for MoE transformers and provides an accompanying training recipe. The routing method (ReMoE) replaces softmax + TopK expert selection with a ReLU activation function. This results in a fully differentiable MoE transformer, which requires a new loss penalty to encourage a balanced load and a reasonable number of active experts ($k$). The authors empirically evaluate their new method in the context of autoregressive language modeling. Experiments include a study of performance w.r.t loss at different model sizes, a varied number of experts, and different expert granularity; an analysis of different training stages; and an analysis of routing behavior.

**Strengths:**

- The paper is generally well-written and easy to understand. In particular, the method section is thorough and gives a comprehensive explanation of the modifications made to the standard MoE training pipeline. I quite like Figure 4 as it gives a precise picture of the training dynamics specific to ReMoEs.
- The proposed method outperforms all other methods compared to in the study.
- Experiments in Figure 6 studying performance improvements for different numbers of active parameters, different expert granularity, and a varying number of experts demonstrate that the proposed ReLU MoEs along with their training algorithm consistently outperforms a standard TopK MoE.
- Using the ReLU activation function as a replacement to TopK + softmax for MoE routing is a novel and potentially interesting idea. The method could allow for improved conditional computation as there is no hard requirement for each token to be processed by exactly $k$ experts.

**Weaknesses:**

My **greatest concern** regards the attribution of the success of the method. As the paper is currently written, it suggests that using the ReLU activation function in place of TopK + softmax is the cause of the improved performance because it makes the routing function *differentiable*. However, the training algorithm is also changed. Notably, during the first ~100 steps of training the ReMoE has sparsity as low as 20% (Figure 4 (a)), requiring substantially more memory and computational cost for these first 100 steps. This leads me to ask the following question:

*Is the success of the method due to the use of ReLU or is it due to the expensive near-dense training of the MoE for the first 100 steps?*

A simple way to address this weakness would be to provide a dMoE baseline that trains with k=int( 0.8 * E ) (e.g., nearly dense) for the first 100 steps of training and switches to k=1 thereafter.


**Other weaknesses**
- While the ReMoE method allows for more flexible routing, it could unevenly distribute the load across a sequence, causing latency issues during autoregressive generation. How does the allocation of compute vary across the sequence of tokens?
- In the introduction, you state "the vanilla TopK router introduces a discrete and nondifferentiable training objective (Shazeer et al., 2017; Zoph et al., 2022), limiting the performance". This is false. The training objective (e.g. auxiliary loss), itself, is differentiable. The difficulty is related to receiving a sparse gradient (e.g., a gradient for only a subset of activated experts).
- Recent relevant works [1-3] are not mentioned in the related work section and not compared to in the main manuscript. Specifically, the sparse-mixer-v2 method, which improves the router's gradient estimate would be a relevant baseline to compare with.
- I miss an evaluation of the performance of ReMoEs on LM evaluation benchmarks.
- A batch size of 512k tokens is small for LLM pre-training.
- While authors claim to train on a compute-optimal number of tokens, is 30B compute optimal for all models in the study? Many recent LLMs train well beyond the compute-optimal regime (e.g. Llama3 8B was trained for more than 50x compute optimal). Do the ReMoE results hold for longer training?
- ReMoEs will result in high memory consumption early on in training. This is reasonable for smaller models, but can quickly become expensive for very sparse MoEs. This should be explicitly noted in section 3.5 line 269.





[1] SPARSE BACKPROPAGATION FOR MOE TRAINING, https://arxiv.org/pdf/2310.00811

[2] GRIN: GRadient-INformed MoE https://arxiv.org/abs/2409.12136

[3] Dense Training, Sparse Inference: Rethinking Training of Mixture-of-Experts Language Models https://arxiv.org/pdf/2404.05567



**I would be happy to raise my score if some concerns are addressed.**

**Questions:**

One of the benefits of ReLU routing is that there is no inherent requirement for each token to be processed by $k$ experts.

- Can ReMoE leverage the flexibility it is said to have with the regularization terms used?





**Suggested writing changes:**
- Scaling in parameters N --> Scaling in active parameters N

---

> ### Author Response · Authors · 2024-11-22
> **Response to Reviewer qrQT (1/3)**
>
> We would like to express our gratitude to the reviewer for recognizing the **novelty** and **potential** of our work, as well as for providing insightful and constructive feedback. We sincerely hope that, after considering our responses, the reviewer will reconsider the score. Below, we address the reviewer’s concerns in detail:
>
> **Q1: Greatest concern on attribution of the success: is it due to the extra cost in the first 100 steps?**
>
> We sincerely thank the reviewer for raising this important question and appreciate the suggestion to evaluate this aspect. However, the first 100 steps are not particularly expensive. Our measurements of the end-to-end time consumption for different stages of training MoE and ReMoE reveal that the overhead of Stage I/II (the first 100 steps) accounts for only **0.58%** of the total training time.
>
> **Table: End-to-End Training Time Comparison of MoE and ReMoE (Unit: Hours).**
>
> | Model | Stage I/II (the first 100 steps) | Stage III | Total  |
> | ----- | ---------- | --------- | ------ |
> | MoE   | 0.53       | 119.12    | 119.65 |
> | ReMoE | 1.23       | 119.25    | 120.48 |
>
>
> Following the reviewer's advice, we constructed a dMoE baseline where training starts with $k = \text{int}(0.8 \cdot E) = 6$ experts out of $E = 8$ for the first 100 steps, then switches to $k = 1$ thereafter. We refer to this configuration as "MoE with warmup." The results of this experiment are as follows:
>
> **Table: Comparison of MoE, MoE with warmup, and ReMoE**
>
> | Model           | Valid Loss | Downstream Avg. Acc. | ARC-c | ARC-e | BoolQ | HellaSwag | LAMBADA | PIQA  | RACE  |
> | --------------- | ---------- | -------------------- | ----- | ----- | ----- | --------- | ------- | ----- | ----- |
> | MoE             | 1.937      | 39.67                | 20.05 | 45.16 | 57.83 | 29.83     | 32.97   | 63.55 | 28.33 |
> | MoE with warmup | 1.928      | 39.29                | 20.73 | 46.38 | 52.35 | 30.28     | 33.90   | 63.76 | 27.66 |
> | ReMoE           | 1.921      | 40.03                | 20.22 | 46.68 | 54.16 | 30.26     | 35.94   | 63.55 | 29.38 |
>
> The results show that "MoE with warmup" achieves a slightly lower validation loss but slightly worse downstream performance compared to the standard MoE. Notably, ReMoE outperforms both. This observation suggests that the first 100 steps, accounting for only 0.16% of the total training procedure, do not provide a critical improvement to overall performance.
>
> To further investigate, we extended the warmup setting to our scaling experiments with larger $E$. In these experiments, we used more experts during Stages I and II, mirroring ReMoE, ensuring that the compute budgets for MoE and ReMoE were equivalent across all three stages:
>
> **Table: Results for MoE with warmup under different expert count $E$**
>
> | Model, E=8      | Valid Loss | Downstream Avg. Acc. | Model, E=32     | Valid Loss | Downstream Avg. Acc. | Model, E=128    | Valid Loss | Downstream Avg. Acc. |
> | --------------- | ---------- | -------------------- | --------------- | ---------- | -------------------- | --------------- | ---------- | -------------------- |
> | MoE             | 1.937      | 39.67                | MoE             | 1.875      | 39.63                | MoE             | 1.855      | 40.89                |
> | MoE with warmup | 1.928      | 39.29                | MoE with warmup | 1.869      | 40.06                | MoE with warmup | 1.841      | 41.34                |
> | ReMoE           | 1.921      | 40.03                | ReMoE           | 1.852      | 41.58                | ReMoE           | 1.815      | 42.12                |
>
> We find that when $E$ increases, the warmup setting does yield slightly larger improvements since more compute is used. But ReMoE still performs better, and the the steeper slope in $E$ preserves.
>
> From another perspective, as shown in Figure 5, the training loss of ReMoE at 25B tokens is comparable to that of dMoE with 30B tokens. This gap is far larger than the overhead in Stage I/II, proving the superiority of ReLU routing.
>
> Due to time constraints, we are unable to apply warmup to all our experiments during the rebuttal phase. Additionally, MoE with warmup is not a standard approach, and its results have not been commonly reported in previous literature. However, for the camera-ready version, we will include updated results with extended training steps for MoE to match the additional compute used in Stages I and II of ReMoE across all scaling experiments.

---

> > ### Author Response · Authors · 2024-11-22
> > **Response to Reviewer qrQT (2/3)**
> >
> > **Q2: Computation allocation across tokens and its latency concern:**
> >
> > Figure 7 illustrates the dynamic computation allocation across different tokens in ReMoE. ReMoE does distribute compute unevenly across a sequence, assigning fewer experts to frequent tokens such as `\n` and `the` while allocating more experts to rarer tokens. Please refer to Section 5.1 for more details.
> >
> > However, it doesn't cause latency issues. The average number of activated experts is constrained by the controlled sparsity, ensuring that the average latency remains consistent with that of MoE.
> >
> > We measured the training and inference throughput of ReMoE and MoE, observing only minor differences. The relative speed of ReMoE compared to MoE is summarized below, where ↑ indicates ReMoE is faster and ↓ indicates it is slower:
> >
> > **Table: relative throughput of ReMoE over MoE**
> >
> > | N    | Training | Inference |
> > | ---- | -------- | --------- |
> > | 182M | ↑1.82%   | ↑2.19%    |
> > | 469M | ↓1.37%   | ↑3.89%    |
> > | 978M | ↓1.77%   | ↓0.23%    |
> >
> > Details regarding the speed measurements can be found in Appendix D.
> >
> >
> > **Q3: Is TopK non-differentiable?**
> >
> > We thank the reviewer for raising the issue of "sparse gradients" in TopK. This actually aligns with what we referred to as "discrete". In fact, the "discontinuous and non-differentiable" statement is commonly adopted by related works including SoftMoE, SMEAR, Lory \[1,2,3\].
> >
> > Technically, TopK is treated as a differentiable operation in `PyTorch Autograd`, with the "derivative" effectively slicing the gradient (i.e., producing a sparse gradient). However, this differentiability only holds _**piecewise**_ (as shown in Figure 2). The training objective (language model loss) is discrete when the activated experts change. For example, consider a simple MoE $f(x)=\tilde\theta_1f_1(x)+\tilde\theta_2f_2(x)$, where $\tilde{\theta_1},\tilde{\theta_2}=\text{Top1}(\theta_1,\theta_2)$. Depending on the comparison, we have $f(x)=\theta_1f_1(x)$ if $\theta_1>\theta_2$ and $f(x)=\theta_2f_2(x)$ if $\theta_1<\theta_2$. The output $f(x)$ is discontinuous and thus non-differentiable at $\theta_1=\theta_2$ since $f_1\ne f_2$.
> >
> > We believe a more precise explanation of the difficulty is: the sparse gradient becomes inaccurate when active experts change. For instance, in the above example when active expert changes from $f_1$ to $f_2$, the sparse gradient computed with $f_1$​ no longer accurately reflects the true gradient for the router, which is now related to $f_2$​. This issue does not arise with ReLU, as its gradient is inherently sparse and aligned with its output.
> >
> > From the perspective of sparse gradients, the gradient of Softmax is dense, but we artificially slice it into a sparse one using TopK. While techniques such as Gumbel Softmax sampling and gradient approximations, as in SparseMixer-v2, allow us to use sparse gradients to approximate dense gradients, they introduce additional variance. In contrast, with ReLU, the router’s output is naturally sparse, and its gradient is also inherently sparse. This allows the router to be trained correctly and efficiently with the sparse gradient, without the need for sampling or approximation.
> >
> > **Q4: Comparison with SparseMixer-v2 and DS-MoE**
> >
> > We appreciate the reviewer for highlighting these relevant works. We agree that SparseMixer-v2, which was introduced in September 2024 and improves TopK's gradient estimation, is highly relevant, although contemporaneous works after July 1, 2024 are not required to be compared according to [ICLR FAQ](https://iclr.cc/Conferences/2025/FAQ).
> >
> > We use the official implementation of SparseMixer-v2 in [this link](https://huggingface.co/microsoft/GRIN-MoE/blob/main/modeling_grinmoe.py) and add it as one of the baselines of our main experiment in Section 4.2. The updated results are presented below:
> >
> > **Table: Comparison of ReMoE with SparseMixer-v2**
> >
> > | Model          | Valid Loss | Downstream Avg. Acc. | ARC-c | ARC-e | BoolQ | HellaSwag | LAMBADA | PIQA  | RACE  |
> > | -------------- | ---------- | -------------------- | ----- | ----- | ----- | --------- | ------- | ----- | ----- |
> > | MoE            | 1.937      | 39.67                | 20.05 | 45.16 | 57.83 | 29.83     | 32.97   | 63.55 | 28.33 |
> > | SparseMixer-v2 | 1.935      | 38.39                | 19.80 | 46.72 | 45.96 | 30.24     | 34.12   | 62.89 | 29.00 |
> > | ReMoE          | 1.921      | 40.03                | 20.22 | 46.68 | 54.16 | 30.26     | 35.94   | 63.55 | 29.38 |
> >
> > We find SparseMixer-v2 achieves a slightly lower valid loss over MoE but still underperforms ReMoE.
> >
> > Regarding DS-MoE, the reported results indicate sparsity levels below 75%. In our experiments with an 87.5% sparsity setting, we observed divergence. Furthermore, DS-MoE introduces a pipeline modification that trains the dense network with all experts active and later applies TopK during inference. This approach is orthogonal to our work, which focuses specifically on improving routing mechanisms.

---

> > > ### Author Response · Authors · 2024-11-22
> > > **Response to Reviewer qrQT (3/3)**
> > >
> > > **Q5: Downstream evaluation results**
> > >
> > > In the revised version, we have included downstream evaluation results for **all experiments**. These results demonstrate that ReMoE consistently outperforms MoE in downstream tasks, aligning with the perplexity results. For a detailed breakdown of the results, please refer to Appendix E.
> > >
> > > **Q6: Small batch size & Limited training tokens**
> > >
> > > The batch size in our experiments aligns with the settings used in Megablocks\[4\]. And 30B tokens are larger than the compute-optimal tokens for all models, including the largest model of 978M active parameters. (~28B predicted in \[5\])
> > >
> > > As for the scalability concerns with larger batch sizes and longer training, we conducted experiments with an extended training duration for the $N = 469$M active parameter models in both MoE and ReMoE. Specifically, we trained for 120B tokens (4x more tokens than our main experiments) using a larger batch size of 4M tokens. The results in the following table confirm that our original conclusions still hold. The detailed results are added to Appendix C.
> > >
> > > **Table: Results for active parameters $N=$ 469M models trained on 120B tokens with batch size of 4M tokens**
> > >
> > > | Model | Valid Loss | Downstream Avg. Acc. | ARC-c | ARC-e | BoolQ | HellaSwag | LAMBADA | PIQA  | RACE  |
> > > | ----- | ---------- | -------------------- | ----- | ----- | ----- | --------- | ------- | ----- | ----- |
> > > | MoE   | 1.716      | 44.12                | 23.62 | 52.40 | 53.94 | 35.43     | 43.64   | 68.34 | 31.48 |
> > > | ReMoE | 1.689      | 45.49                | 25.34 | 55.22 | 55.96 | 36.76     | 45.82   | 68.93 | 30.43 |
> > >
> > >
> > > **Q7: High memory consumption in the beginning**
> > >
> > > We appreciate the reviewer for highlighting the non-negligible memory issue. At the start of ReMoE training, the memory required for parameters and optimizer states is equivalent to that of MoE. However, the memory usage for activations is indeed higher. This is because ReMoE activates more experts than MoE, and the intermediate activations of all activated experts must be saved for the backward pass.
> > >
> > > This issue can be significantly mitigated using the **activation checkpointing** technique, which avoids storing intermediate results of activated experts by recomputing them on-the-fly during the backward pass. With activation checkpointing, the memory overhead is reduced proportionally to the number of layers.
> > >
> > > We measured the memory usage of MoE and ReMoE with $N = 469$M, $E = 8$, using activation checkpointing and a micro-batch size of 8. The peak memory usage shows a modest increase of **2.8%** for ReMoE compared to MoE (28.59GB->29.40GB).
> > >
> > > Besides, in practice when memory is constrained, a smaller micro-batch size can be used during Stages I and II—which account for only a small proportion of the training process—followed by a larger micro-batch size comparable to MoE during Stage III, where the memory consumption of ReMoE is identical to that of MoE.
> > >
> > > The statement of high memory consumption and its solution is added at the end of Section 3.5.
> > >
> > >
> > >
> > > **Q8: Dynamic expert allocation with regularization**
> > >
> > > Figure 7 illustrates the dynamic expert allocation under the refined load-balancing regularization in ReMoE. Notably, the regularization in ReMoE targets only those experts receiving an excessive number of tokens, without restricting the assignment of tokens to different number of experts. The plot highlights that ReMoE efficiently assigns different compute to different tokens, with the allocation correlated to token frequency: rarer tokens are allocated more computational resources.
> > >
> > > **Q9: Writing changes**
> > >
> > > Thank you for the constructive feedback! We have changed "Scaling in Parameters $N$" to "Scaling in Active Parameters $N$" for improved clarity and accuracy.
> > >
> > >
> > > \[1\] From Sparse to Soft Mixtures of Experts https://arxiv.org/pdf/2308.00951
> > >
> > > \[2\] Soft Merging of Experts with Adaptive Routing https://arxiv.org/pdf/2306.03745
> > >
> > > \[3\] Lory: Fully Differentiable Mixture-of-Experts for Autoregressive Language Model Pre-training https://arxiv.org/pdf/2405.03133
> > >
> > > \[4\] MegaBlocks: Efficient Sparse Training with Mixture-of-Experts https://arxiv.org/pdf/2211.15841
> > >
> > > \[5\] Scaling Laws for Fine-Grained Mixture of Experts https://arxiv.org/pdf/2402.07871

---

> > > > ### Comment · Reviewer_qrQT · 2024-11-25
> > > > **[reviewer reply to Q2-Q9] My concerns have been addressed, I am raising my score.**
> > > >
> > > > The authors' answers to Q2-9 are satisfactory to me. I particularly appreciate the effort to include SparseMixer-v2; I think this considerably strengthens the paper.
> > > >
> > > > With respect to Q7, you make a good point that increasing the number of gradient accumulation steps during this phase can allow near-dense training during this phase without needing to re-shard the checkpoint (should be mentioned in the manuscript). However, I don't think this will be the case for all model configurations.

---

> > > > > ### Author Response · Authors · 2024-11-27
> > > > > **Thank You for Raising the Score**
> > > > >
> > > > > We sincerely thank the reviewer for the thoughtful reconsideration and for raising the score. We greatly value your time and effort in providing constructive feedback to help us improve our work. Below, we have carefully addressed the follow-up questions:
> > > > >
> > > > > **Q10: Training curves for TopK with warmup vs ReMoE**
> > > > >
> > > > > Thank the reviewer again for the insightful suggestion regarding TopK-routed MoE with warmup. In response, we have conducted additional experiments and included the detailed results in Appendix H. The comparison of training curves is now presented in Figure 13. We observe that the training curve of TopK with warmup runs parallel to those of TopK and ReLU, showing a subtle improvement over TopK while underperforming compared to ReLU.
> > > > >
> > > > > Previously, this analysis was not included in the paper as we prioritized conducting further experiments to refine all scaling results before the camera-ready version. We agree with your perspective that the experiments would be more equitable if the compute cost is matched rather than the training steps.
> > > > >
> > > > >
> > > > >
> > > > > **Q11: Performance of TopK with warmup when warmup steps increase**
> > > > >
> > > > > We conducted additional experiments by increasing the near-dense warmup steps for TopK MoE from 100 to 500 and 1000, starting with $k=6$ and transitioning to $k=1$. Interestingly, we observed that increasing the number of warmup steps resulted in **worse** performance.
> > > > >
> > > > > This outcome is attributed to the discrepancy between the training objectives of $k=6$ (warmup phase) and $k=1$ (post-warmup phase). A closer inspection of the loss transition before and after warmup reveals that when warmup ends at 100 steps, the gap is relatively small, with the loss changing from 6.491 to 6.751. However, with extended warmup steps of 500 and 1000, the gap becomes significantly larger, shifting from 3.101 to 5.827 and from 2.695 to 4.428, respectively.
> > > > >
> > > > > We further experimented with 50 warmup steps and observed a relatively higher validation loss compared to 100 steps. This outcome is expected, as it reflects the reduced computation during the warmup phase. These results indicate that the optimal number of warmup steps is approximately 100.
> > > > >
> > > > > **Table: Valid loss of MoE with different warmup steps**
> > > > >
> > > > > | warmup steps | 0     | 50    | 100   | 500   | 1000  |
> > > > > | ------------ | ----- | ----- | ----- | ----- | ----- |
> > > > > | Valid Loss   | 1.937 | 1.930 | 1.928 | 1.930 | 1.931 |
> > > > >
> > > > >
> > > > > The **key takeaways** are as follows:
> > > > >
> > > > > - Near-dense warmup can enhance the performance of TopK when training from scratch by providing a better initialization for the experts.
> > > > > - The warmup phase should conclude while the language model loss is still decreasing rapidly; otherwise, a significant gap between the warmup and subsequent training phases may arise, leading to a drop in overall performance.
> > > > >
> > > > > From this perspective, ReMoE can naturally identify the appropriate number of warmup steps (approximately 100–150 steps in our setting, as suggested in Appendix B for various $\lambda_0$ values) and the optimal warmup sparsity (as demonstrated by the self-adaptive sparsity curve in Stage I/II described in Section 3.5). This capability is attributed to the continuity and differentiability of the ReLU router.
> > > > >
> > > > > We have included this discussion in detail in Appendix H.
> > > > >
> > > > > **Q12: Further memory concern regarding the need for re-sharding checkpoints**
> > > > >
> > > > > We thank the reviewer for raising this practical question. To clarify, the memory overhead associated with training the near-dense model at the beginning involves **only** the additional **activation memory** required to store intermediate results of the activated experts.
> > > > >
> > > > > We also appreciate the reviewer’s acknowledgment of our solution to increase gradient accumulation steps as a means to avoid re-sharding checkpoints. This discussion has been added to Section 3.5. However, we agree that this approach may not be suitable for all model configurations, particularly for extremely large models where the micro-batch size per device is already 1.
> > > > >
> > > > > In such cases, activation checkpointing can be employed as an alternative. This approach avoids storing the intermediate results of all active experts by recomputing them on-the-fly during the backward pass, significantly reducing activation memory usage and eliminating the need for re-sharding checkpoints. While activation checkpointing introduces additional forward-pass computations, its overall impact on training is negligible, as it would only be utilized during a small fraction of the training steps.

---

> ### Author Response · Authors · 2024-11-25
> **Looking Forward to Your Feedback**
>
> Dear Reviewer qrQT:
>
> We really appreciate your insightful and constructive comments that help us improve this paper. We have taken our maximum effort to address all your concerns.
>
> As the ICLR public discussion phase concludes in two days, we kindly ask if our responses have fully addressed your concerns. If there are any remaining issues, we’d be happy to provide further clarifications. If you feel that your concerns have been resolved, we would greatly appreciate it if you could consider raising the score.
>
> Thank you once again for your time and thoughtful input. We hope you have a wonderful day!
>
> Best wishes,
>
> The Authors

---

> ### Comment · Reviewer_qrQT · 2024-11-25
> **Reply to Q1 answer**
>
> Thank you for taking the time to run these experiments.
>
> **Cost of the first 100 steps** I agree with the author's point that 0.58% of total training time is negligible. For longer industry-scale training runs this number will also shrink. The only obstacle I see to adoption, here, are logistical issues related to the need for re-sharding checkpoints if significant increases in parallelism are required to fit the *temporarily more dense* model in memory (this should be the case for the largest dense models). Such issues are addressable, however, and should not stand in the way of publication.
>
> **Attribution of success** I read the new results as follows: 1) while appreciated for reproducibility, downstream average accuracy is noisy and unreliable at this model size, 2) validation accuracy can be relied upon at this model scale, and 3) it seems that warmup does help the TopK MoE improve over the baseline, but the improvement relative to ReLU MoE diminishes with more experts. This addresses my concern that the improvement of ReLU MoE is entirely due to the near-dense phase at the beginning of training. If longer near-dense training allows TopK to match the performance of ReLU MoE, the authors can still claim their methods require less near-dense training.
>
>
> **Follow-up questions/comments**
> - How do the training curves look for TopK with warmup vs ReMoE? You included plots for other experiments during the rebuttal phase why not these?
> - If the number of near-dense training steps for TopK increases say from 100 to 500 or 1000 does the final performance of the TopK MoE also improve?

---

### Official Review · Reviewer_VcWp · 2024-11-04

**Soundness:** 3
**Presentation:** 3
**Contribution:** 2
**Rating:** 6
**Confidence:** 4

**Summary:**

This work proposed a new MoE architecture that replace the TopK with ReLU in the routing modules. The ReLU-MoE is then optimized in a three stage training framework, experiments demonstrates the effectiveness of the proposed method across language modeling on different model sizes.

**Strengths:**

- Various ablation studies and visualization are conducted, analysing the effectivenss of the proposed methods.

- The method is clearly explained and easy to follow.

**Weaknesses:**

- The novelty of ReMoE is modest, as it simply replaces the TopK operation with ReLU.

- The perplexity improvements over dMoE are not significantly. whether downstream performance, such as on common-sense reasoning tasks, would shows significant enhancement?

- In practical MoE implementations, an all-to-all dispatch and combine approach can be used to assign each token to the appropriate experts, thereby reducing memory usage and computational load. However, during the initial stage of training, the absence of high sparsity can impact memory and computation efficiency, as this phase effectively requires training a nearly dense model.

**Questions:**

- Why ReLU based router can enhance the domain specialization of MoE?

- Whether the ReLU-based router can be further enhanced with shifted-ReLU, even with a learnable threshold?

- How does the stage I/II affects the training performance?

---

> ### Author Response · Authors · 2024-11-22
> **Response to Reviewer VcWp (1/2)**
>
> We thank the reviewer for finding our method **clear** and **easy to follow**. We value the constructive feedback and have carefully considered the concerns raised. Below we provide detailed responses to address the reviewer's concerns:
>
> **Q1: Lack of Novelty?**
>
> - While replacing TopK+Softmax with ReLU may appear simple, the underlying reasons for its effectiveness are nontrivial: This approach addresses the issue of discontinuity in TopK-routed MoE and facilitates dynamic expert allocation.
> - Additionally, controlling the sparsity of the ReLU router and integrating load balancing mechanisms pose nontrivial challenges that we successfully address.
> - Notably, ReMoE is **the first** fully differentiable MoE architecture applicable to autoregressive models that outperforms TopK-routed MoE.
>
> **Q2: Improvement Not Significant?**
>
> While the improvement in final validation loss may seem small in absolute terms (~0.8% decrease from 1.937 to 1.921 in Figure 6(a)), it is significant in three ways:
> - **Token efficiency:** The perplexity result translates to approximately **16.7%** fewer tokens required to achieve the same perplexity: ReMoE needs only about 25B tokens to reach a validation loss of 1.937, compared to 30B tokens for dMoE.
> - **Expert efficiency:** As shown in Figure 6(b), ReMoE with **32** experts outperforms MoE with **128** experts.
> - **Scalability:** ReMoE exhibits a steeper scaling slope with expert counts ($E$), suggesting it offers greater potential for improvement as $E$ increases compared to MoE.
>
> **Q3: Downstream evaluation results**
>
> In the revised version, we have included downstream evaluation results for **all experiments**. These results demonstrate that ReMoE consistently outperforms MoE in downstream tasks, aligning with the perplexity results. For a detailed breakdown of the results, please refer to Appendix E.
>
> **Q4: Effect of stage I/II on efficiency, memory and performance**
>
> - Efficiency: Stages I and II indeed activate more experts and require additional compute. However, they comprise only ~100 steps, which is approximately **0.17%** of the total training process. As such, the time overhead is negligible. To provide further clarity, we included an end-to-end training time comparison in Appendix D. The results indicate that the time overhead from Stages I and II amounts to just **0.58%**.
>
> **Table: End-to-End Training Time Comparison of MoE and ReMoE (Unit: Hours)**
>
> | Model | Stage I/II | Stage III | Total  |
> | ----- | ---------- | --------- | ------ |
> | MoE   | 0.53       | 119.12    | 119.65 |
> | ReMoE | 1.23       | 119.25    | 120.48 |
>
> - Memory: Stages I and II activate more experts compared to MoE, resulting in higher activation memory consumption. However, this issue can be significantly mitigated using the **activation checkpointing** technique, which avoids storing intermediate results of extra activated experts by recomputing them on-the-fly during the backward pass. With activation checkpointing, the memory overhead is reduced proportionally to the number of layers.
>
>   To quantify this, we measured the memory usage of MoE and ReMoE with $N = 469$M, $E = 8$, using activation checkpointing and a micro batch size of 8. The peak memory usage shows a modest increase of **2.8%** for ReMoE compared to MoE (28.59GB->29.40GB).
>
>   Clarification of the memory overhead is made in Section 3.5.
>
> - Performance: The two stages utilize more compute and thus slightly increase the performance. But the improvement is marginal since they account for only 0.17% of the total training process. To further investigate, we conducted an experiment by training a TopK MoE with a similar warmup strategy: activating nearly all experts during the first 100 steps and switching to $k = 1$ thereafter. The results show that MoE with warmup achieves a slightly lower validation loss but slightly worse downstream performance compared to standard MoE. Both configurations underperform relative to ReMoE:
>
> **Table: Comparison of MoE, MoE with warmup (extra compute in Stage I/II), and ReMoE**
>
> | Model           | Valid Loss | Downstream Avg. Acc. | ARC-c | ARC-e | BoolQ | HellaSwag | LAMBADA | PIQA  | RACE  |
> | --------------- | ---------- | -------------------- | ----- | ----- | ----- | --------- | ------- | ----- | ----- |
> | MoE             | 1.937      | 39.67                | 20.05 | 45.16 | 57.83 | 29.83     | 32.97   | 63.55 | 28.33 |
> | MoE with warmup | 1.928      | 39.29                | 20.73 | 46.38 | 52.35 | 30.28     | 33.90   | 63.76 | 27.66 |
> | ReMoE           | 1.921      | 40.03                | 20.22 | 46.68 | 54.16 | 30.26     | 35.94   | 63.55 | 29.38 |

---

> > ### Author Response · Authors · 2024-11-22
> > **Response to Reviewer VcWp (2/2)**
> >
> > **Q5: Why ReLU enhance domain specialization**
> >
> > Domain specialization is a phenomenon that is also observed in other differentiable MoE variants like Lory\[1\]. However, a comprehensive theoretical explanation remains an open question.
> >
> > Our hypothesis is that:
> > - In TopK-routed MoE, the discontinuity in switching active experts forces the experts to be more similar to prevent rapid loss growth during transitions. In contrast, ReMoE’s differentiability allows for a more diverse expert distribution, facilitating domain specialization.
> >
> > - Dynamic allocation in ReMoE also fosters domain specialization. Our analysis in Appendix G of activated expert counts across different domains reveals clear variations, highlighting a connection between dynamic allocation and domain specialization. This behavior may arise because certain easy tokens require fewer than $k$ experts, but TopK forces them to utilize $k$ experts, resulting in different experts being trained on similar combinations of tokens.
> >
> > **Table: Average active expert count of ReMoE on different domains:**
> >
> > | Domain | Layer 0 | Layer 5 | Layer 11 | Avg.   |
> > | ------ | ------- | ------- | -------- | ------ |
> > | Arxiv  | 0.2789  | 1.4121  | 1.6358   | 1.1262 |
> > | Books  | 0.2266  | 1.3927  | 2.6665   | 1.2942 |
> > | C4     | 0.2515  | 1.5505  | 2.8319   | 1.3208 |
> > | Github | 0.1996  | 0.9584  | 0.8938   | 0.8335 |
> > | Stack  | 0.3090  | 1.1061  | 0.9691   | 0.9453 |
> > | Wiki   | 0.3759  | 1.1288  | 0.6687   | 0.6776 |
> >
> >
> > **Q6: Possible improvements of shifted-ReLU**
> >
> > Thank you for the insightful and constructive suggestion! In fact, we did explore shifted-ReLU in our early experiments, incorporating a learnable threshold as a method to control sparsity. However, we found that it was not effective in boosting sparsity during training.
> >
> > To evaluate this approach, we conducted an experiment using learnable shifted-ReLU, starting with a 200-step warmup phase followed by a zeroth-order algorithm to update the shift value. The results of this experiment are summarized below:
> >
> > **Table: Sparsity control results for shifted-ReLU with learnable shift value.**
> >
> > | Steps       | 200   | 300   | 400   | 500   | 600   | ... | 20000 |
> > | ----------- | ----- | ----- | ----- | ----- | ----- | --- | ----- |
> > | Shift value | 0     | 0.097 | 0.195 | 0.336 | 0.5   | ... | 0.5   |
> > | Sparsity    | 0.774 | 0.812 | 0.816 | 0.809 | 0.798 | ... | 0.752 |
> >
> > As observed, the shift value steadily increases, yet the sparsity unexpectedly decreases after 400 steps and consistently remains below the target value of 0.875. The shift value eventually stabilizes at 0.5 due to additive roundoff errors.
> > We attribute this issue to the following reasons:
> > - *Interaction between router weights and shift value:* During training, the router weights adjust simultaneously with the shift value. This results in larger output values, effectively canceling the sparsity-inducing effect of the shift.
> > - *Numerical precision with bfloat16:* Our pretraining uses the mainstream bfloat16 format. When employing shifted-ReLU, where the router output is centered around a value greater than 0, the representable numbers in this range are less dense, leading to larger roundoff errors.
> >
> > Due to these limitations, we chose to use ReLU combined with L1 regularization as our final method.
> >
> >
> >
> > \[1\] Lory: Fully Differentiable Mixture-of-Experts for Autoregressive Language Model Pre-training https://arxiv.org/pdf/2405.03133

---

> ### Author Response · Authors · 2024-11-25
> **Looking Forward to Your Feedback**
>
> Dear Reviewer VcWp:
>
> We really appreciate your insightful and constructive comments that help us improve this paper. We have taken our maximum effort to address all your concerns.
>
> As the ICLR public discussion phase concludes in two days, we kindly ask if our responses have fully addressed your concerns. If there are any remaining issues, we’d be happy to provide further clarifications. If you feel that your concerns have been resolved, we would greatly appreciate it if you could consider raising the score.
>
> Thank you once again for your time and thoughtful input. We hope you have a wonderful day!
>
> Best wishes,
>
> The Authors

---

> > ### Comment · Reviewer_VcWp · 2024-11-27
> >
> > Thanks for the responses. My concerns have been addressed!

---

> > > ### Author Response · Authors · 2024-11-27
> > > **Thank You for Raising the Score**
> > >
> > > Dear Reviewer VcWp,
> > >
> > > We sincerely thank you for raising the score! We believe that ReLU-routed ReMoE represents a meaningful progression from the current discontinuous TopK-routed MoE and hope it inspires future research on fully differentiable Mixture-of-Experts.
> > >
> > > Best regards,
> > >
> > > The Authors

---

### Meta-Review · Area_Chair_LZ6W · 2024-12-13

**Metareview:**

The paper introduces a novel approach to address key limitations of traditional TopK-routed MoE architectures by replacing the discrete TopK routing mechanism with a ReLU-based differentiable router. The authors present a compelling case for the ReLU router, highlighting its ability to ensure smooth gradients, dynamic expert allocation, and efficient load balancing. Their experiments demonstrate significant improvements over traditional MoE architectures, both in terms of perplexity and downstream task performance, across various model sizes, expert counts, and token granularities. This work also introduces practical enhancements, such as a three-stage training process that stabilizes the ReLU routing, which, despite its simplicity, yields promising results. The authors provide extensive empirical validation, including ablation studies and comparisons with state-of-the-art baselines like SparseMixer-v2.

While some reviewers noted that the novelty of replacing TopK with ReLU might seem modest at first glance, the authors convincingly argue that the continuous nature of ReLU enables dynamic computation scaling and a steeper scalability curve for expert counts, making it a meaningful advancement for MoE models. The paper’s clear presentation and rigorous rebuttal process, where all major concerns were satisfactorily addressed, reflect its readiness for publication.

**Additional Comments On Reviewer Discussion:**

Reviewers raised concerns about the attribution of improvements to the ReLU router, the computational overhead of the initial training stages, the lack of downstream evaluation results, and the clarity of the proposed regularization mechanism. The authors addressed these comprehensively by providing additional experiments, including comparisons with nearly dense TopK baselines, extended downstream evaluations, and insights into the interplay of ReLU's continuity and dynamic expert allocation. They also clarified memory and computational trade-offs, demonstrating minimal overhead from the ReLU router's initialization phase. These responses resolved key concerns and reinforced the paper’s strengths

---

### Decision · Program_Chairs · 2025-01-22

Accept (Poster)